# Redeeming Intrinsic Rewards via Constrained Optimization

**Eric Chen**,[*] **Zhang-Wei Hong** [*†], **Joni Pajarinen**[‡] & **Pulkit Agrawal**[†§]
Improbable AI Lab, Massachusetts Institute of Technology
MIT-IBM Watson AI Lab[†]      Aalto University[‡]
NSF AI Institute for AI and Fundamental Interactions (IAIFI)[§]

## Abstract

State-of-the-art reinforcement learning (RL) algorithms typically use random sampling (e.g., $\epsilon$-greedy) for exploration, but this method fails on hard exploration tasks like Montezuma's Revenge. To address the challenge of exploration, prior works incentivize exploration by rewarding the agent when it visits novel states. Such *intrinsic* rewards (also called exploration bonus or curiosity) often lead to excellent performance on hard exploration tasks. However, on easy exploration tasks, the agent gets distracted by intrinsic rewards and performs unnecessary exploration even when sufficient task (also called extrinsic) reward is available. Consequently, such an overly curious agent performs worse than an agent trained with only task reward. Such inconsistency in performance across tasks prevents the widespread use of intrinsic rewards with RL algorithms. We propose a principled constrained optimization procedure called *Extrinsic-Intrinsic Policy Optimization* (EIPO) that automatically tunes the importance of the intrinsic reward: it suppresses the intrinsic reward when exploration is unnecessary and increases it when exploration is required. The results is superior exploration that does not require manual tuning in balancing the intrinsic reward against the task reward. Consistent performance gains across sixty-one ATARI games validate our claim. The code is available at https://github.com/Improbable-AI/eipo.

## 1   Introduction

The goal of reinforcement learning (RL) [1] is to find a mapping from states to actions (i.e., a policy) that maximizes reward. At every learning iteration, an agent is faced with a question: has the maximum possible reward been achieved? In many practical problems, the maximum achievable reward is unknown. Even when the maximum achievable reward is known, if the current policy is sub-optimal then the agent is faced with another question: would spending time improving its current strategy lead to higher rewards (*exploitation*), or should it attempt a different strategy in the hope of discovering potentially higher reward (*exploration*)? Pre-mature *exploitation* is akin to getting stuck in a *local-optima* and precludes the agent from exploring. Too much exploration on the other hand can be distracting, and prevent the agent from perfecting a good strategy. Resolving the *exploration-exploitation* dilemma [1] is therefore essential for data/time efficient policy learning.

In simple decision making problems where actions do not affect the state (e.g. bandits or contextual bandits [2]), provably optimal algorithms for balancing exploration against exploitation are known [3, 2]. However, in the general settings where RL is used, such algorithms are unknown. In the absence of methods that work well in both theory and practice, state-of-the-art RL algorithms rely on heuristic exploration strategies such as adding noise to actions or random sampling of sub-optimal actions

---

[*] denotes equal contribution. Correspondence to `pulkitag@mit.edu`.

36th Conference on Neural Information Processing Systems (NeurIPS 2022).

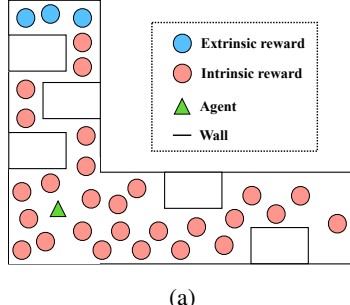
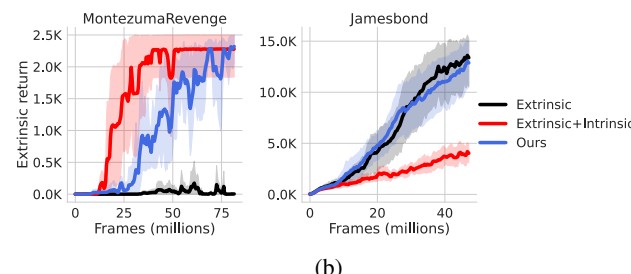

(a)                                                    (b)

Figure 1: **(a)** At the start of training all locations are novel for the agent (green triangle), and therefore the pink circles representing *intrinsic rewards* are evenly distributed across the map. The blue circles represent sources of *extrinsic reward* or task-reward. Here *intrinsic* rewards can distract the agent, as the sum of *extrinsic* and *intrinsic* rewards can be increased by moving along the bottom corridor. **(b)** This type of distraction is a possible reason why an intrinsic reward method does not consistently outperform a method trained using only extrinsic rewards across ATARI games. Intrinsic rewards help in some games where the task or extrinsic reward is sparse (e.g., *Montezuma's revenge*), but hurt in other games such as *James Bond*. Our proposed method, EIPO, intelligently uses intrinsic rewards when needed and consistently matches the best-performing algorithm amongst extrinsic and extrinsic+intrinsic methods.

(e.g., $\epsilon$-greedy). However, such strategies fail in sparse reward scenarios where infrequent rewards hinder policy improvement. One such task is the notorious ATARI game, *Montezuma's Revenge* [4].

Sparse reward problems can be solved by supplementing the task reward (or extrinsic reward $r_E$) with a dense exploration bonus (or intrinsic reward $r_I$) generated by the agent itself [4–8]. Intrinsic rewards encourage the agent to visit novel states, which increases the chance of encountering states with task reward. Many prior works [4, 8, 9] show that jointly optimizing for intrinsic and extrinsic reward (i.e., $r_E + \lambda r_I$, where $\lambda \geq 0$ is a hyperparameter) instead of only optimizing for extrinsic reward (i.e., $\lambda = 0$) improves performance on sparse reward tasks such as *Montezuma's revenge* [4, 9, 10].

However, a recent study found that using intrinsic rewards does not consistently outperform simple exploration strategies such as $\epsilon$-greedy across ATARI games [11]. This is because the mixed objective $(r_E + \lambda r_I)$ is *biased* for $|\lambda| > 0$, and optimizing it does not necessarily yield the optimal policy with respect to the extrinsic reward alone [12]. Fig. 1a illustrates this problem using a toy example. Here the green triangle is the agent and the blue/pink circles denote the location of extrinsic and intrinsic rewards, respectively. At the start of training, all states are novel and provide a source of intrinsic reward (i.e., pink circles). This makes accumulating intrinsic rewards easy, which the agent may exploit to optimize its objective of maximizing the sum of intrinsic and extrinsic rewards. However, such optimization can result in a local maxima: the agent might move rightwards along the bottom corridor, essentially distracting the agent from the blue task rewards at the top. In this example, since it is not hard to find the task reward, better performance is obtained if only the extrinsic reward $(\lambda = 0)$ is maximized. The trouble, however, is that in most environments one doesn't know a priori how to optimally trade off intrinsic and extrinsic rewards (i.e., choose $\lambda$).

A common practice is to conduct an extensive hyperparameter search to find the best $\lambda$, as different values of $\lambda$ are best suited for different tasks (see Fig. 4). Furthermore, as the agent progresses on a task, the best exploration-exploitation trade-off can vary, and a constant $\lambda$ may not be optimal throughout training. In initial stages of training exploration might be preferred. Once the agent is able to obtain some task reward, it might prefer exploiting these rewards instead of exploring further. The exact dynamics of the exploration-exploitation trade-off is task-dependent, and per-task tuning is tedious, undesirable, and often computationally infeasible. Consequently, prior works use a fixed $\lambda$ during training, which our experiments reveal is sub-optimal.

We present an optimization strategy that alleviates the need to manually tune the relative importance of extrinsic and intrinsic rewards as training progresses. Our method leverages the *bias* of intrinsic rewards when it is useful for exploration and mitigates this bias when it does not help accumulate higher extrinsic rewards. This is achieved using an *extrinsic optimality constraint* that forces the extrinsic rewards earned after optimizing the mixed objective to be equal to the extrinsic rewards accumulated by the *optimal* policy that maximizes extrinsic rewards only. Enforcing the *extrinsic optimality constraint* in general settings is intractable because the optimal extrinsic reward is unknown. We devise a practical algorithm called **Extrinsic-Intrinsic Policy Optimization (EIPO)**, which uses an approximation to solve this constrained optimization problem (Section 3).

While in principle we can apply EIPO to any intrinsic reward method, we evaluate performance using state-of-the-art *random network distillation (RND)* [9]. Fig. 1b presents *teaser* results on two ATARI games: (i) *Montezuma's Revenge* - where joint optimization with RND (red) substantially outperforms a PPO policy [13] optimized with only extrinsic rewards (black); (ii) *James Bond* - where PPO substantially outperforms RND. These results reinforce the notion that bias introduced by intrinsic rewards helps in some games, but hurts in others. Our algorithm EIPO (blue) matches the best algorithm in both games, showing that it can leverage intrinsic rewards as needed. Results across 61 ATARI games reinforce this finding. Additionally, in some games EIPO outperforms multiple strong baselines with and without intrinsic rewards, indicating that our method can not only mitigate the potential performance decreases caused by intrinsic reward bias, but can also improve performance beyond the current state-of-the-art.

## 2 Preliminaries

We consider a discrete-time Markov Decision Process (MDP) consisting of a state space $\mathcal{S}$, an action space $\mathcal{A}$, and an extrinsic reward function $\mathcal{R}_E : \mathcal{S} \times \mathcal{A} \to \mathbb{R}$. We distinguish the extrinsic and intrinsic reward components by $E$ and $I$, respectively. The extrinsic reward function $\mathcal{R}_E$ refers to the actual task objective (e.g., game score). The agent starts from an initial state $s_0$ sampled from the initial state distribution $\rho_0 : \mathcal{S} \to \mathbb{R}$. At each timestep $t$, the agent perceives a state $s_t$ from the environment, takes action $a_t$ sampled from the policy $\pi$, receives extrinsic reward $r_t^E = \mathcal{R}_E(s_t, a_t)$, and moves to the next state $s_{t+1}$ according to the transition function $\mathcal{T}(s_{t+1}|s_t, a_t)$. The agent's goal is to use interactions with the environment to find the optimal policy $\pi$ such that the extrinsic objective value $J_E(\pi)$ is maximized:

$$\max_{\pi} J_E(\pi), \text{ where } J_E(\pi) = \mathbb{E}_{s_0, a_0, \cdots \sim \pi} \Big[ \sum_{t=0}^{\infty} \gamma^t r_t^E \Big] \text{ (Extrinsic objective)}, \tag{1}$$

$$s_0 \sim \rho_0, a_t \sim \pi(a|s_t), s_{t+1} \sim \mathcal{T}(s_{t+1}|s_t, a_t) \, \forall t > 0$$

where $\gamma$ denotes a discount factor. For brevity, we abbreviate $\mathbb{E}_{s_0, a_0, \cdots \sim \pi} \Big[ . \Big]$ as $\mathbb{E}_{\pi} \Big[ . \Big]$ unless specified.

Intrinsic reward based exploration strategies[4, 8, 9] attempt to encourage exploration by providing "intrinsic rewards" (or "exploration bonuses") that incentivize the agent to visit unseen states. Using the intrinsic reward function $\mathcal{R}_I : \mathcal{S} \times \mathcal{A} \to \mathbb{R}$, the optimization objective becomes:

$$\max_{\pi \in \Pi} J_{E+I}(\pi), \text{ where } J_{E+I}(\pi) = \mathbb{E}_{\pi} \Big[ \sum_{t=0}^{\infty} \gamma^t (r_t^E + \lambda r_t^I) \Big] \text{ (Mixed objective)}, \tag{2}$$

where $\lambda$ denotes the intrinsic reward scaling coefficient. We abbreviate the intrinsic reward at timestep $t$ as $r_t^I = \mathcal{R}_I(s_t, a_t)$. State-of-the-art intrinsic reward based exploration strategies [9, 14] often optimize the objective in Eq. 2 using Proximal Policy Optimization (PPO) [13].

## 3 Mitigating the Bias of Intrinsic Rewards

Simply maximizing the sum of intrinsic and extrinsic rewards does not guarantee a policy that also maximizes extrinsic rewards: $\arg\max_{\pi_{E+I} \in \Pi} J_{E+I}(\pi_{E+I}) \neq \arg\max_{\pi_E \in \Pi} J_E(\pi_E)$. At convergence the optimal policy $\pi_{E+I}^* = \arg\max_{\pi_{E+I}} J_{E+I}(\pi_{E+I})$ could be suboptimal w.r.t. $J_E$, which measures the agent's task performance. Because the agent's performance is measured using extrinsic reward only, we propose enforcing an *extrinsic optimality constraint* that ensures the optimal "mixed" policy $\pi_{E+I}^* = \arg\max_{\pi_{E+I}} J_{E+I}(\pi_{E+I})$ leads to as much extrinsic reward as the optimal "extrinsic" policy $\pi_E^* = \arg\max_{\pi_E \in \Pi} J_E(\pi_E)$. The resulting optimization objective is:

$$\max_{\pi_{E+I} \in \Pi} J_{E+I}(\pi_{E+I}) \tag{3}$$

$$\text{subject to } J_E(\pi_{E+I}) - \max_{\pi_E} J_E(\pi_E) = 0 \qquad \text{(Extrinsic optimality constraint)}.$$

Solving this optimization problem can be viewed as proposing a policy $\pi_{E+I}$ that maximizes $J_{E+I}$, and then checking if the proposed $\pi_{E+I}$ is feasible given the extrinsic optimality constraint.

The constrained objective is difficult to optimize because evaluating the extrinsic optimality constraint requires $J_E(\pi_E^*)$, which is unknown. To solve this optimization problem, we transform it into an *unconstrained min-max* optimization problem using Lagrangian duality (Section 3.1). We then describe an iterative algorithm for solving the min-max optimization by alternating between minimization and maximization in Section 3.2, and we present implementation details in Section 3.3.

## 3.1 The Dual Objective: Unconstrained Min-Max Optimization Problem

The Lagrangian dual problem for the primal constrained optimization problem in Eq. 3 is:

$$\min_{\alpha \in \mathbb{R}^+} \left[ \max_{\pi_{E+I} \in \Pi} J_{E+I}(\pi_{E+I}) + \alpha \big( J_E(\pi_{E+I}) - \max_{\pi_E \in \Pi} J_E(\pi_E) \big) \right], \tag{4}$$

where $\alpha \in \mathbb{R}^+$ is the Lagrangian multiplier. We rewrite Eq. 4 by merging $J_{E+I}(\pi)$ and $J_E(\pi)$:

$$J_{E+I}^\alpha(\pi_{E+I}) := J_{E+I}(\pi_{E+I}) + \alpha J_E(\pi_{E+I}) = \mathbb{E}_{\pi_{E+I}} \left[ \sum_{t=0}^\infty \gamma^t \big[ (1+\alpha) r^E(s_t, a_t) + r^I(s_t, a_t) \big] \right].$$

The re-written objective provides an intuitive interpretation of $\alpha$: larger values correspond to increasing the impetus on extrinsic rewards (i.e., exploitation). Substituting $J_{E+I}^\alpha(\pi_{E+I})$ into Eq. 4 and rearranging terms yields the following min-max problem:

$$\min_{\alpha \in \mathbb{R}^+} \left[ \max_{\pi_{E+I} \in \Pi} \min_{\pi_E \in \Pi} J_{E+I}^\alpha(\pi_{E+I}) - \alpha J_E(\pi_E) \right]. \tag{5}$$

## 3.2 Algorithm for Optimizing the Dual Objective

We now describe an algorithm for solving $\pi_E$, $\pi_{E+I}$, and $\alpha$ in each of the sub-problems in Eq. 5.

**Extrinsic policy $\pi_E$ (min-stage).** $\pi_E$ is optimized via the minimization sub-problem, which can be re-written as a maximization problem:

$$\min_{\pi_E \in \Pi} J_{E+I}^\alpha(\pi_{E+I}) - \alpha J_E(\pi_E) \quad \rightarrow \quad \max_{\pi_E \in \Pi} \alpha J_E(\pi_E) - J_{E+I}^\alpha(\pi_{E+I}) \tag{6}$$

The main challenge is that evaluating the objectives $J_{E+I}^\alpha(\pi_{E+I})$ and $J_E(\pi_E)$ requires sampling trajectories from both policies $\pi_{E+I}$ and $\pi_E$. If one were to use an on-policy optimization method such as PPO, this would require sampling trajectories from two separate policies at each iteration during training, which would be data inefficient. Instead, if we assume that the two policies are similar, then we can leverage results from prior work to use the trajectories from one policy ($\pi_{E+I}$) to *approximate* the return of the other policy ($\pi_E$) [15, 16, 13].

First, using the performance difference lemma from [15], the objective $\alpha J_E(\pi_E) - J_{E+I}^\alpha(\pi_{E+I})$ can be re-written (see Appendix A.1.3 for detailed derivation):

$$\alpha J_E(\pi_E) - J_{E+I}^\alpha(\pi_{E+I}) = \mathbb{E}_{\pi_E} \left[ \sum_{t=0}^\infty \gamma^t U_{\min}^{\pi_{E+I}}(s_t, a_t) \right]. \tag{7}$$

$$\text{where } U_{\min}^{\pi_{E+I}}(s_t, a_t) := \alpha r_t^E + \gamma V_{E+I}^{\pi_{E+I}}(s_{t+1}) - V_{E+I}^{\pi_{E+I}}(s_t),$$

$$V_{E+I}^{\pi_{E+I}}(s_t) := \mathbb{E}_{\pi_{E+I}} \left[ \sum_{t=0}^\infty \gamma^t (r_t^E + r_t^I) | s_0 = s_t \right]$$

Next, under the *similarity assumption*, a lower bound to the objective in Eq. 7 can be obtained [13]:

$$\mathbb{E}_{\pi_E} \left[ \sum_{t=0}^\infty \gamma^t U_{\min}^{\pi_{E+I}}(s_t, a_t) \right] \geq \mathbb{E}_{\pi_{E+I}} \left[ \sum_{t=0}^\infty \gamma^t \min \left\{ \frac{\pi_E(a_t|s_t)}{\pi_{E+I}(a_t|s_t)} U_{\min}^{\pi_{E+I}}(s_t, a_t), \right. \right.$$
$$\left. \left. \text{clip} \left( \frac{\pi_E(a_t|s_t)}{\pi_{E+I}(a_t|s_t)}, 1 - \epsilon, 1 + \epsilon \right) U_{\min}^{\pi_{E+I}}(s_t, a_t) \right\} \right] \tag{8}$$

where $\epsilon \in [0, 1]$ denotes a threshold. Intuitively, this clipped objective (Eq. 8) penalizes the policy $\pi_E$ that behaves differently from $\pi_{E+I}$ because overly large or small $\frac{\pi_E(a_t|s_t)}{\pi_{E+I}(a_t|s_t)}$ terms are clipped. More details are provided in Appendix A.2.

**Mixed policy $\pi_{E+I}$ (max-stage).** The sub-problem of solving for $\pi_{E+I}$ is posed as

$$\max_{\pi_{E+I}\in\Pi} J^\alpha_{E+I}(\pi_{E+I}) - \alpha J_E(\pi_E). \tag{9}$$

We again rely on the approximation from [13] to derive a lower bound surrogate objective for $J^\alpha_{E+I}(\pi_{E+I}) - \alpha J_E(\pi_E)$ as follows (see Appendix A.1.2 for details):

$$
\begin{aligned}
J^\alpha_{E+I}(\pi_{E+I}) - \alpha J_E(\pi_E) \geq \mathbb{E}_{\pi_E}\Big[ \sum_{t=0}^{\infty} \gamma^t \min\Big\{ &\frac{\pi_{E+I}(a_t|s_t)}{\pi_E(a_t|s_t)} U^{\pi_E}_{\max}(s_t, a_t), \\
&\text{clip}\left( \frac{\pi_{E+I}(a_t|s_t)}{\pi_E(a_t|s_t)}, 1-\epsilon, 1+\epsilon \right) U^{\pi_E}_{\max}(s_t, a_t)\Big\}\Big],
\end{aligned} \tag{10}
$$

where $U^{\pi_E}_{\max}$ and $V^{\pi_E}_E$ are defined as follows:

$$U^{\pi_E}_{\max}(s_t, a_t) := (1+\alpha)r^E_t + r^I_t + \gamma\alpha V^{\pi_E}_E(s_{t+1}) - \alpha V^{\pi_E}_E(s_t)$$

$$V^{\pi_E}_E(s_t) := \mathbb{E}_{\pi_E}\Big[ \sum_{t=0}^{\infty} \gamma^t r^E_t | s_0 = s_t \Big].$$

**Lagrangian multiplier $\alpha$.** We solve for $\alpha$ by using gradient descent on the surrogate objective derived above. Let $g(\alpha) := \max_{\pi_{E+I}\in\Pi} \min_{\pi_E\in\Pi} J^\alpha_{E+I}(\pi_{E+I}) - \alpha J_E(\pi_E)$. Therefore, $\nabla g(\alpha) = J_E(\pi_{E+I}) - J_E(\pi_E)$. We approximate $\nabla g(\alpha)$ using the lower bound surrogate objective:

$$
\begin{aligned}
J_E(\pi_{E+I}) - J_E(\pi_E) \geq L(\pi_E, \pi_{E+I}) = \mathbb{E}_{\pi_E}\Big[ \sum_{t=0}^{\infty} \gamma^t \min\Big\{ &\frac{\pi_{E+I}(a_t|s_t)}{\pi_E(a_t|s_t)} A^{\pi_E}(s_t, a_t), \\
&\text{clip}\left( \frac{\pi_{E+I}(a_t|s_t)}{\pi_E(a_t|s_t)}, 1-\epsilon, 1+\epsilon \right) A^{\pi_E}(s_t, a_t)\Big\}\Big],
\end{aligned} \tag{11}
$$

where $A^{\pi_E}(s_t, a_t) = r^E_t + \gamma V^{\pi_E}_E(s_{t+1}) - V^{\pi_E}_E(s_t)$ is the advantage of taking action $a_t$ at state $s_t$, and then following $\pi_E$ for subsequent steps. We update $\alpha$ using a step size $\beta$ (Appendix A.1.4):

$$\alpha \leftarrow \alpha - \beta L(\pi_E, \pi_{E+I}). \tag{12}$$

Unlike prior works that use a fix trade off between extrinsic and intrinsic rewards, optimizing $\alpha$ during training allows our method to automatically and dynamically tune the trade off.

### 3.3 Implementation

**Min-max alternation schedule.** We use an iterative optimization scheme that alternates between solving for $\pi_E$ by minimizing the objective in Eq. 6, and solving for $\pi_{E+I}$ by maximizing the objective in Eq. 9 (`max_stage`). We found that switching the optimization every iteration hinders performance which is hypothesize is a result of insufficient training of each policy. Instead, we switch between the two stages when the objective value does not improve anymore. Let $J[i]$ be the objective value $J^\alpha_{E+I}(\pi_{E+I}) - \alpha J_E(\pi_E)$ at iteration $i$. The switching rule is:

$$
\begin{cases}
J[i] - J[i-1] \leq 0 \implies \text{max\_stage} \leftarrow \text{False}, & \text{if max\_stage} = \text{True} \\
J[i] - J[i-1] \geq 0 \implies \text{max\_stage} \leftarrow \text{True}, & \text{if max\_stage} = \text{False},
\end{cases}
$$

where `max_stage` is a binary variable indicating whether the current stage is the max-stage. $\alpha$ is only updated when the max-stage is done using Eq. 12.

**Parameter sharing.** The policies $\pi_{E+I}$ and $\pi_E$ are parametrized by two separate multi-layer perceptrons (MLP) with a shared CNN backbone. The value functions $V^{\pi_{E+I}}_{E+I}$ and $V^{\pi_E}_E$ are represented in the same fashion, and share a CNN backbone with the policy networks. When working with image inputs (e.g., ATARI), sharing the convolutional neural network (CNN) backbone between $\pi_E$ and $\pi_{E+I}$ helps save memory, which is important when using GPUs (in our case, an NVIDIA RTX 3090Ti). If both policies share a backbone however, maximizing the objective (Eq. 9) with respect to $\pi_{E+I}$ might interfere with $J_E(\pi_E)$, and impede the performance of $\pi_E$. Similarly, minimizing the objective (Eq. 6) with respect to $\pi_E$ might modify $J^\alpha_{E+I}$ and degrade the performance of $\pi_{E+I}$.

**Algorithm 1** Extrinsic-Intrinsic Policy Optimization (EIPO)

---

1: Initialize policies $\pi_{E+I}$ and $\pi_E$, `max_stage[0]` $\leftarrow$ `False`, and $J[0] \leftarrow 0$
2: **for** $i = 1 \cdots$ **do**                                         $\triangleright$ $i$ denotes iteration index
3:     **if** `max_stage[i - 1]` **then**                  $\triangleright$ Max-stage: rollout by $\pi_E$ and update $\pi_{E+I}$
4:         Collect trajectories $\tau_E$ using $\pi_E$ and compute $U_{\max}^{\pi_E}(s_t, a_t)$ $\forall (s_t, a_t) \in \tau_E$
5:         Update $\pi_{E+I}$ by Eq. 10 and $\pi_E$ by auxiliary objective (Section 3.3)
6:         $J[i] \leftarrow J_{E+I}^{\alpha}(\pi_{E+I}) - \alpha J_E(\pi_E)$
7:         `max_stage[i]` $\leftarrow J[i] - J[i-1] \leq 0$
8:     **else**                                              $\triangleright$ Min-stage: rollout by $\pi_{E+I}$ and update $\pi_E$
9:         Collect trajectories $\tau_{E+I}$ using $\pi_{E+I}$ and compute $U_{\min}^{\pi_{E+I}}(s_t, a_t)$ $\forall (s_t, a_t) \in \tau_{E+I}$
10:         Update $\pi_E$ by Eq. 8 and $\pi_{E+I}$ by auxiliary objective (Section 3.3)
11:         $J[i] \leftarrow J_{E+I}^{\alpha}(\pi_{E+I}) - \alpha J_E(\pi_E)$
12:         `max_stage[i]` $\leftarrow J[i] - J[i-1] \geq 0$
13:     **end if**
14:     **if** `max_stage[i - 1]` = `True` and `max_stage[i]` = `False` **then**
15:         Update $\alpha$ (Eq. 32)                        $\triangleright$ Update when the max-stage is done
16:     **end if**
17: **end for**

---

Interference can be avoided by introducing auxiliary objectives that prevent decreases in $J_E(\pi_E)$ and $J_{E+I}^{\alpha}$ when optimizing for $\pi_{E+I}, \pi_E$, respectively. For example, when updating $\pi_{E+I}$ in the max-stage, $\max_{\pi_{E+I}} J_{E+I}^{\alpha}(\pi_{E+I}) - \alpha J_E(\pi_E)$, the auxiliary objective $\max_{\pi_E} J_E(\pi_E)$ can prevent updates in the shared CNN backbone from decreasing $J(\pi_E)$. We incorporate the auxiliary objective without additional hyperparameters (Appendix A.2.3). The overall objective is in Appendix A.2.4.

*Extrinsic-Intrinsic Policy Optimization (EIPO)* - the policy optimization algorithm we introduce to solve Eq. 5, is outlined in Algorithm 1. Pseudo-code can be found in Algorithm 2, and full implementation details including hyperparameters can be found in Appendix A.2.

## 4   Experiments

While EIPO is agnostic to the choice of intrinsic rewards, we mainly experiment with RND [9] because it is the state-of-the-art intrinsic reward method. EIPO implemented with RND is termed *EIPO-RND* and compared against several baselines below. All policies are learned using PPO [13].

- **EO (Extrinsic only)**: The policy is trained using extrinsic rewards only: $\pi^* = \arg\max_{\pi \in \Pi} J_E(\pi)$.

- **RND (Random Network Distillation) [9]**: The policy is trained using the sum of extrinsic rewards $(r_t^E)$ and RND intrinsic rewards $(\lambda r_t^I)$: $\pi^* = \arg\max_{\pi \in \Pi} J_{E+I}(\pi)$. For ATARI experiments, we chose a single value of $\lambda$ that worked best across games. Other methods below also use this $\lambda$.

- **EN (Ext-norm-RND)**: We found that a variant of RND where the extrinsic rewards are normalized using running mean and standard deviation (Appendix A.2.2) outperforms the original RND implementation, especially in games where RND performs worse than EO. Our method EIPO and other baselines below are therefore implemented using *Ext-norm-RND*.

- **DY (Decay-RND)**: Instead of having a fixed trade-off between exploration and exploitation throughout training, dynamically adjusting exploration vs. exploitation can lead to better performance. Without theoretical results describing how to make such adjustments, a commonly used heuristic is to gradually transition from exploration to exploitation. One example is $\epsilon$-greedy exploration [17], where $\epsilon$ is decayed over the course of training. Similarly, we propose a variant of *Ext-norm-RND* where intrinsic rewards are progressively scaled down to eliminate exploration bias over time. Intrinsic rewards $r_t^I$ are scaled by $\lambda(i)$, where $i$ denotes the iteration number. The objective function turns into $J_{E+I} = \mathbb{E}_{\pi}\left[\sum_{t=0}^{\infty} \gamma^t (r_t^E + \lambda(i)r_t^I)\right]$, where $\lambda(i)$ is defined as $\lambda(i) = \text{clip}(\frac{i}{I}(\lambda_{\max} - \lambda_{\min}), \lambda_{\min}, \lambda_{\max})$. Here, $\lambda_{\max}$ and $\lambda_{\min}$ denote the predefined maximum and minimum $\lambda(i)$, and $I$ is the iteration after which decay is fixed. We follow the linear decay schedule used to decay $\epsilon$-greedy exploration in DQN [17].

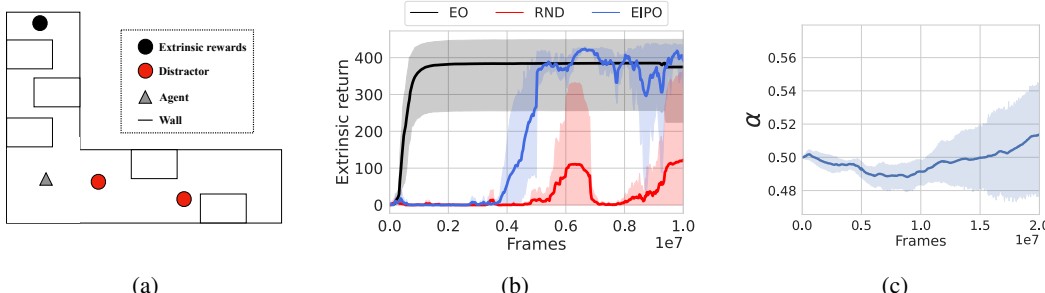

(a)                                      (b)                                      (c)

Figure 2: **(a)** 2D navigation task with a similar distribution of extrinsic and intrinsic rewards as Figure 1a. The gray triangle is the agent. The **black** dot is the goal that provides extrinsic reward. Red dots are randomly placed along the bottom corridor at the start of every episode. Due to the novelty of their locations, these dots serve as a source of intrinsic rewards. **(b)** RND is distracted by the intrinsic rewards and fails to match EO optimized with only extrinsic rewards. Without intrinsic rewards, EO is inferior to our method EIPO in terms of discovering a good path to the black goal. This result indicates that EIPO is not distracted by intrinsic rewards, and uses them as necessary to improve performance. **(c)** $\alpha$ controls the importance of the extrinsic-intrinsic optimality constraint. It decreases until the extrinsic return starts to rise between $0.5$ and $1.0$ million frames. Afterwards, $\alpha$ also increases, showing that our method emphasizes intrinsic rewards at the start of training, and capitalizes on extrinsic rewards once they are found.

- **DC (Decoupled-RND) [18]**: The primary assumption in EIPO is that $\pi_{E+I}$ and $\pi_E$ are similar (Section 3.2). A recent work used this assumption in a different way to balance intrinsic and extrinsic rewards [18]: they regularize the mixed policy to stay close to the extrinsic policy by minimizing the Kullback–Leibler (KL) divergence $D_{KL}(\pi_E||\pi_{E+I})$. However, they do not impose the *extrinsic-optimality constraint*, which is a key component in EIPO. Therefore, comparing against this method called *Decoupled-RND* (*DC*) [18] will help separate the gains in performance that come from making the *similarity assumption* versus the importance of *extrinsic-optimality constraint*. We adapted *DC* to make use of *Ext-norm-RND* intrinsic rewards:

$$\pi_{E+I}^* = \arg\max_{\pi_{E+I}} \mathbb{E}_{\pi_{E+I}} \Big[ \sum_{t=0}^{\infty} \gamma^t (r_t^E + r_t^I) - D_{KL}(\pi_E||\pi_{E+I}) \Big], \quad \pi_E^* = \arg\max_{\pi_E} \mathbb{E}_{\pi_I} \Big[ \sum_{t=0}^{\infty} \gamma^t r_t^E \Big].$$

### 4.1 Illustrative example

We exemplify the problem of intrinsic reward bias using a simple 2D navigation environment (Fig. 2a) implemented using the Pycolab game engine [19]. The gray sprite denotes the agent, which observes a $5 \times 5$ pixel view of its surroundings. For every timestep the agent spends at the (**black**) location at the top of the map, an extrinsic reward of +1 is provided. The red circles are randomly placed in the bottom corridor at the start of each episode. Because these circles randomly change location, they induce RND intrinsic rewards throughout training. Because intrinsic rewards along the bottom corridor are easier to obtain than the extrinsic reward, optimizing the mixed objective $J_{E+I}$ can yield a policy that results in the agent exploring the bottom corridor (i.e., exploiting the intrinsic reward) without ever discovering the extrinsic reward at the top. Fig. 2b plots the evolution of the average extrinsic return during training for EO, RND, and *EIPO-RND* across 5 random seeds. We find that *EIPO-RND* outperforms both RND and EO. RND gets distracted by the intrinsic rewards (red blocks) and performs worse than the agent that optimizes only the extrinsic reward (EO). EO performs slightly worse than *EIPO-RND*, possibly because in some runs the EO agent fails to reach the goal without the guidance of intrinsic rewards.

To understand why *EIPO-RND* performs better, we plot the evolution of the parameter $\alpha$ that trades-off exploration against exploitation during training (Fig. 2c). Lower values of $\alpha$ denote that the agent is prioritizing intrinsic rewards (exploration) over extrinsic rewards (exploitation; see Section 3.1). This plot shows that for the first $\sim 0.5M$ steps, the value of $\alpha$ decreases (Fig. 2c) indicating that the agent is prioritizing *exploration*. Once the agent finds the extrinsic reward (between $0.5M - 1M$ steps), the value of $\alpha$ stabilizes which indicates that further prioritization of *exploration* is unnecessary. After $\sim 1M$ steps the value of $\alpha$ increases as the agent prioritizes *exploitation*, and extrinsic return increases (Fig. 2b). These plots show that EIPO is able to dynamically trade-off exploration against exploitation during training. The dynamics of $\alpha$ during training also support the intuition that EIPO transitions from exploration to exploitation over the course of training.

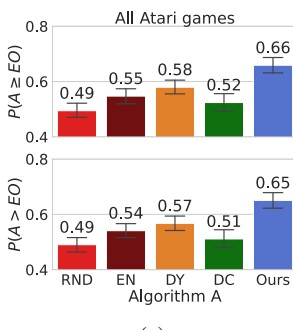 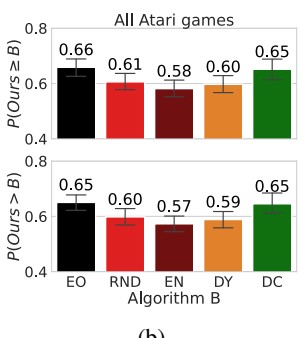 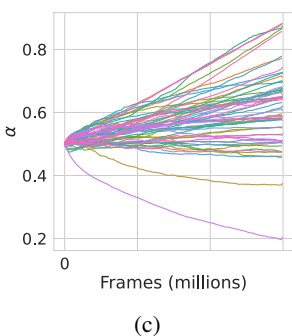

|     |     |     |
| --- | --- | --- |
| (a) | (b) | (c) |

Figure 3: **(a)** *EIPO-RND* (ours) has a higher probability of improvement $P(EIPO\text{-}RND > \mathrm{EO})$ over EO than all other baselines. This suggests *EIPO-RND* is more likely to attain a higher score than EO, compared to other methods. **(b)** Probability of improvement $P(EIPO\text{-}RND > B) > 0.5\ \forall B$ indicates that *EIPO-RND* performs strictly better than the baselines in the majority of trials. **(c)** Each colored curve denotes the evolution of $\alpha$ in a specific game using EIPO. The variance in $\alpha$ trajectories across games reveals that different exploration-exploitation dynamics are best suited for different games.

## 4.2 EIPO Redeems Intrinsic Rewards

To investigate if the benefits of EIPO carry over to more realistic settings, we conducted experiments on ATARI games [20], the de-facto benchmark for exploration methods [4, 11]. Our goal is not to maximize performance on ATARI, but to use ATARI as a proxy to anticipate performance of different RL algorithms on a new and potentially real-world task. Because ATARI games vary in terms of task objective and exploration difficulty, if an algorithm $A$ *consistently* outperforms another algorithm $B$ on ATARI, it provides evidence that $A$ may outperform $B$ given a new task with unknown exploration difficulty and task objective. If such an algorithm is found, the burden of cycling through exploration algorithms to find the one best suited for a new task is alleviated.

We used a "probability of improvement" metric $P(A > B)$ with a 95%-confidence interval (see Appendix A.4.2; [21]) to judge if algorithm $A$ consistently outperforms $B$ across all ATARI games. If $P(A > B) > 0.5$, it indicates that algorithm $A$ outperforms $B$ on a majority of games (i.e., *consistent* improvement). Higher values of $P(A > B)$ means greater consistency in performance improvement of $A$ over $B$. For the sake of completeness, we also report $P(A \geq B)$ which represents a weaker improvement metric that measures whether algorithm $A$ matches or outperforms algorithm $B$. These statistics are calculated using the median of extrinsic returns over the last hundred episodes for at least 5 random seeds for each method and game. More details are provided in Appendix A.4.

**Comparing Exploration with Intrinsic Rewards v/s Only Optimizing Extrinsic Rewards** Results in Fig. 3a show that $P(\mathrm{RND} > \mathrm{EO}) = 0.49$ with confidence interval $\{0.46, 0.52\}$, where EO denotes PPO optimized using extrinsic rewards only. These results indicate inconsistent performance gains of RND over EO, an observation also made in prior works [11]. *Ext-norm-RND* (*EN*) fares slightly better against EO, suggesting that normalizing extrinsic rewards helps the joint optimization of intrinsic and extrinsic rewards. We find that $P(EIPO\text{-}RND > EO) = 0.65$ with confidence interval $\{0.62, 0.67\}$, indicating that EIPO is able to successfully leverage intrinsic rewards for exploration and is likely to outperform EO on new tasks. Like EIPO, *Decoupled-RND* assumes that $\pi_E$ and $\pi_{E+I}$ are similar, but *Decoupled-RND* is not any better than *Ext-norm-RND* when compared against EO. This suggests that the similarity assumption on its own does not improve performance, and that the *extrinsic-optimality-constraint* plays a key part in improving performance.

**EIPO *Strictly* Outperforms Baseline Methods** Results in Fig. 3b show that $P(EIPO\text{-}RND > B) > 0.5$ across ATARI games in a statistically rigorous manner for all baseline algorithms $B$. Experiments on the Open AI Gym MuJoCo benchmark follow the same trend: EIPO either matches or outperforms the best algorithm amongst EO and RND (see Appendix 10). These results show that EIPO is also applicable to tasks with continuous state and action spaces. Taken together, the results in ATARI and MuJoCo benchmarks provide strong evidence that given a new task, EIPO has the highest chance of achieving the best performance.

## 4.3 Does EIPO Outperform RND Tuned with Hyperparameter Search?

In results so far, RND used the intrinsic reward scaling coefficient $\lambda$ that performed best across games. Our claim is that EIPO can automatically tune the weighting between intrinsic and extrinsic

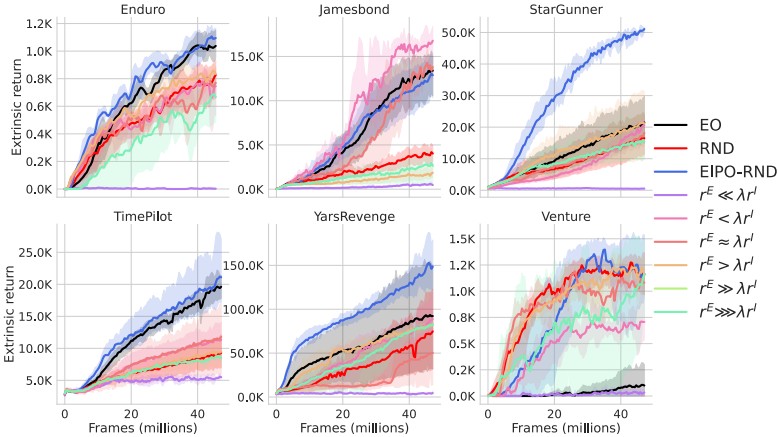

Figure 4: No choice of the intrinsic reward scaling coefficients $\lambda$ consistently yield performance improvements over RND and EO, whereas our method *EIPO-RND* consistently outperforms RND and EO without any tuning across environments. This indicates that our method is less susceptible to $\lambda$.

rewards. If this is indeed true, then EIPO should either match or outperform the version of RND that uses the best $\lambda$ determined on a per-game basis through an exhaustive hyperparameter sweep. Since an exhaustive sweep in each game is time consuming, we evaluate our hypothesis on a subset of representative games: a few games where PPO optimized with only extrinsic rewards (EO) outperforms RND, and a few games where RND outperforms EO (e.g., *Venture* and *Montezuma's Revenge*).

For each game we searched over multiple values of $\lambda$ based on the relative difference in magnitudes between intrinsic ($r^I$) and extrinsic ($r^E$) rewards (see Appendix A.4.4 for details). The results summarized in Fig. 4 show that no choice of $\lambda$ consistently improves RND performance. For instance, $r^E \approx \lambda r^I$ substantially improves RND performance in *Jamesbond*, yet deteriorates performance in *YarsRevenge*. We found that with RND, the relationship between the choice of $\lambda$ and final performance is not intuitive. For example, although *Jamesbond* is regarded as an easy exploration game [4], the agent that uses high intrinsic reward (i.e., $r^E < \lambda r^I$) outperforms the agent with low intrinsic reward ($r^E \gg \lambda r^I$). EIPO overcomes these challenges and is able to automatically adjust the relative importance of intrinsic rewards. It outperforms RND despite game-specific $\lambda$ tuning.

**Dynamic adjustment of the Exploration-Exploitation trade-off during training improves performance**   If the performance gain of EIPO solely resulted from automatically determining a constant trade off between intrinsic and extrinsic rewards throughout training on a per-game basis, an exhaustive hyperparameter search of $\lambda$ should suffice to match the performance of *EIPO-RND*. The fact that *EIPO-RND* outperforms RND with different choices of $\lambda$ leads us to hypothesize that adjusting the importance of intrinsic rewards over the course of training is important for performance. This dynamic adjustment is controlled by the value of $\alpha$ which is optimized during training (Section 3.2). Fig. 3c shows the trajectory of $\alpha$ during training across games. In most games $\alpha$ increases over time, indicating that the agent transitions from exploration to exploitation as training progresses. This decay in exploration is critical for performance: the algorithm, *Decay-RND* ($DY$) that implements this intuition by uniformly *decaying* the importance of intrinsic rewards outperforms RND (Fig. 5 and Fig. 3a). However, in some games $\alpha$ decreases over time indicating that more exploration is required at the end of training. An example is the game *Carnival* where *EIPO-RND* shows a jump in performance at the end of training, which is accompanied by a decrease in $\alpha$ (see Appendix A.8). These observations suggest that simple strategies that follow a fixed schedule to adjust the balance between intrinsic and extrinsic rewards across tasks will perform worse than EIPO. The result that $P(EIPO\text{-}RND > DY) = 0.59$ reported in Fig. 3b confirms our belief and establishes that dynamic, game specific tuning of the exploration-exploitation trade-off is necessary for good performance.

### 4.4   Does EIPO improve over RND only in Easy Exploration Tasks?

The observation that *EIPO-RND* outperforms RND on most games might result from performance improvements on the majority of easy exploration games, at the expense of worse performance on the minority of hard exploration games. To mitigate this possibility, we evaluated performance on games where RND outperforms EO as a proxy for hard exploration. The PPO–normalized scores [21] and 95% confidence intervals of various methods are reported in Table 1. *EIPO-RND* achieves slightly

higher mean score than *Ext-norm-RND*, but lies within the confidence interval. This result suggests that EIPO not only mitigates the bias introduced by intrinsic rewards in easy exploration tasks, but either matches or improves performance in hard exploration tasks.

## 5 Discussion

**EIPO can be a drop-in replacement for any RL algorithm.** Due to inconsistent performance across tasks and difficulties tuning intrinsic reward scaling, bonus based exploration has not been a part of standard state-of-the-art RL pipelines. *EIPO-RND* mitigates these challenges, and outperforms noisy networks [22] - an exploration method that does not use intrinsic rewards, but represents the best performing exploration strategy across ATARI games when used with Q-learning [17] (see Appendix A.9). The consistent performance gains of *EIPO-RND* on ATARI and

Table 1: *EIPO-RND* is either at-part or outperforms RND in the games where RND is better than EO (i.e., hard-exploration tasks).

| Algorithm | PPO-normalized score Mean (CI) | |
|---|---|---|
| RND | 384.57 | (85.57, 756.69) |
| Ext-norm RND | 427.08 | (86.53, 851.52) |
| Decay-RND | 383.83 | (84.19, 753.17) |
| Decoupled-RND | 1.54 | (1.09, 2.12) |
| EIPO-RND | **435.56** | (109.45, 874.88) |

MuJoCo benchmarks make it a strong contender to become the defacto RL exploration paradigm going forward. Though we performed experiments with PPO, the EIPO objective (Eq. 3.1) is agnostic to the particular choice of RL algorithm. As such, it can be used as a drop-in replacement in any RL pipeline.

**Limitations.** Across all 61 ATARI games, we use the same initial value of $\alpha$ and demonstrate that per-task tuning of intrinsic reward importance can be avoided by optimizing $\alpha$. However, it is possible that the initial value of $\alpha$ and learning step-size $\beta$ depend on the choice of intrinsic reward function. To see if this is the case, we tested EIPO with another intrinsic reward metric (ICM [8]) using the same initial $\alpha$ and step-size $\beta$. The preliminary results in Appendix A.10 show that performance gains of *EIPO-ICM* over the baselines are less consistent than *EIPO-RND*. While one possibility is that *RND* is a better intrinsic reward than ICM, the other possibility is that close dependencies between EIPO hyperparameters and the choice of intrinsic reward function come into play. We leave this analysis for future work. Finally, it's worth noting that the performance benefits of EIPO are limited by the quality of the underlying intrinsic reward function on individual tasks. Finding better intrinsic rewards remains an exciting avenue of research, which we hope can be accelerated by removing the need for manual tuning using EIPO.

**Potential applications to reward shaping.** Intrinsic rewards can be thought of as a specific instance of reward shaping [23] – the practice of incorporating auxiliary reward terms to boost overall task performance, which is key to the success of RL in many applications (see examples in [24, 25]). Balancing auxiliary reward terms against the task reward is tedious and often necessitates extensive hyperparameter sweeps. Because the EIPO formulation makes no assumptions specific to intrinsic rewards, its success in balancing the RND auxiliary reward suggests it might also be applicable in other reward shaping scenarios.

## Acknowledgments

We thank members of the Improbable AI Lab for helpful discussions and feedback. We are grateful to MIT Supercloud and the Lincoln Laboratory Supercomputing Center for providing HPC resources. This research was supported in part by the MIT-IBM Watson AI Lab, an AWS MLRA research grant, Google cloud credits provided as part of Google-MIT support, DARPA Machine Common Sense Program, ARO MURI under Grant Number W911NF-21-1-0328, ONR MURI under Grant Number N00014-22-1-2740, and by the United States Air Force Artificial Intelligence Accelerator under Cooperative Agreement Number FA8750-19-2-1000. The views and conclusions contained in this document are those of the authors and should not be interpreted as representing the official policies, either expressed or implied, of the Army Research Office or the United States Air Force or the U.S. Government. The U.S. Government is authorized to reproduce and distribute reprints for Government purposes notwithstanding any copyright notation herein.

## Author Contributions

- **Eric Chen** developed heuristic-based methods for balancing intrinsic and extrinsic rewards which led to the intuition for formulating the extrinsic-intrinsic optimality constraint. He provided critical input during EIPO prototyping. Implemented baselines, ran experiments at scale, and helped with paper writing.

- **Zhang-Wei Hong** conceived of the extrinsic-intrinsic optimality constraint, derived and developed the first prototype of EIPO, ran small-scale experiments, played primary role in paper writing and advised Eric.

- **Joni Pajarinen** provided feedback on the technical correctness of the EIPO formulation and writing.

- **Pulkit Agrawal** conceived and oversaw the project. He was involved in the technical formulation, research discussions, paper writing, overall advising and positioning of the work.

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
