# A  Appendix

 ## A.1  Full derivation

 We present the complete derivation of the objective function in each subproblem defined in Section
 3.2. We start by clarifying notation:

 **Notations**

- $J_E(\pi) = \mathbb{E}_{s_0,a_0,\cdots\sim\pi}\left[\sum_{t=0}^{\infty}\gamma^t r_t^E\right], s_0 \sim \rho_0, a_t \sim \pi(a|s_t), s_{t+1} \sim \mathcal{T}(s_{t+1}|s_t,a_t)\ \forall t > 0$

- $J_{E+I}(\pi) = \mathbb{E}_\pi\left[\sum_{t=0}^{\infty}\gamma^t(r_t^E + r_t^I)\right]$

- $V_E^\pi(s_t) := \mathbb{E}_\pi\left[\sum_{t=0}^{\infty}\gamma^t r_t^E|s_0 = s_t\right]$

- $V_{E+I}^\pi(s_t) := \mathbb{E}_\pi\left[\sum_{t=0}^{\infty}\gamma^t(r_t^E + r_t^I)|s_0 = s_t\right]$

- $V_{E+I}^{\pi,\alpha}(s_t) := \mathbb{E}_\pi\left[\sum_{t=0}^{\infty}\gamma^t((1+\alpha)r_t^E + r_t^I)|s_0 = s_t\right]$

 **Max-stage objective** $U_{\max}^{\pi_E}$.  We show that the objective (LHS) can be approximated by the RHS
 shown below

$$\max_{\pi_{E+I}} J_{E+I}^\alpha(\pi_{E+I}) - \alpha J_E(\pi_E) \approx \max_{\pi_{E+I}} \mathbb{E}_{\pi_E}\left[\frac{\pi_{E+I}(a|s)}{\pi_E(a|s)}U_{\max}^{\pi_E}(s,a)\right] \tag{12}$$
$$\text{subject to } \mathbb{E}_{s\sim\pi_E}\left[D_{KL}(\pi_E(.|s)||\pi_{E+I}(.|s))\right] \leq \delta,$$

 where $\delta$ denotes a constant KL-divergence threshold. We can then expand the LHS as the follows:

$$J_{E+I}^\alpha(\pi_{E+I}) - \alpha J_E(\pi_E)$$
$$= -\alpha J_E(\pi_E) + J_{E+I}^\alpha(\pi_{E+I})$$
$$= -\alpha\mathbb{E}_{s_0\sim\rho_0}\left[\sum_{t=0}^{\infty}\gamma^t V_{\pi_E}^E(s_t)\right] + \mathbb{E}_{\pi_{E+I}}\left[\sum_{t=0}^{\infty}\gamma^t((1+\alpha)r_t^E + r_t^I)\right]$$
$$= \mathbb{E}_{\pi_{E+I}}\left[-\alpha V_E^{\pi_E}(s_0) + \sum_{t=0}^{\infty}\gamma^t((1+\alpha)r_t^E + r_t^I)\right]$$

 For brevity, let $r_t = (1+\alpha)r_t^E + r_t^I$ and $\alpha V_E^{\pi_E}(s_t) = V_t$. Expanding the left-hand of Eq. 12:

$$-V_0 + \sum_{t=0}^{\infty}\gamma^t r_t = (r_0 + \gamma V_1 - V_0) + \gamma(r_1 + \gamma V_2 - V_1) + \gamma^2(r_2 + \gamma V_3 - V_2) + \cdots$$
$$= \sum_{t=0}^{\infty}\gamma^t(r_t + \gamma V_{t+1} - V_t)$$
$$= \sum_{t=0}^{\infty}\gamma^t((1+\alpha)r_t^E + r_t^I + \gamma\alpha V_E^{\pi_E}(s_{t+1}) - \alpha V^{\pi_E}(s_t))$$
$$= \sum_{t=0}^{\infty}\gamma^t U_{\max}^{\pi_E}(s_t,a_t)$$

To facilitate the following derivation, we rewrite the objective $J_{E+I}^{\alpha}(\pi_{E+I}) - \alpha J_E(\pi_E)$:

$$J_{E+I}^{\alpha}(\pi_{E+I}) - \alpha J_E(\pi_E) = \mathbb{E}_{\pi_{E+I}}\left[\sum_{t=0}^{\infty} \gamma U_{\max}^{\pi_E}(s_t, a_t)\right] \tag{13}$$

$$= \sum_{t=0}^{\infty} \sum_{s \in \mathcal{S}} \gamma P(s_t = s | \rho_0, \pi_{E+I}) \sum_{a \in \mathcal{A}} \pi_{E+I}(a|s) U_{\max}^{\pi_E}(s_t, a_t) \tag{14}$$

$$= \sum_{s \in \mathcal{S}} d_{\rho_0}^{\pi_{E+I}, \gamma}(s) \sum_{a \in \mathcal{A}} \pi_{E+I}(a|s) U_{\max}^{\pi_E}(s_t, a_t) \tag{15}$$

$$= \mathbb{E}_{\pi_{E+I}}\left[U_{\max}^{\pi_E}(s_t, a_t)\right] \tag{16}$$

where $d_{\rho_0}^{\pi_{E+I}, \gamma}$ is the discounted state visitation frequency of policy $\pi_{E+I}$ with the initial state distribution $\rho_0$ and discount factor $\gamma$, defined as:

$$d_{\rho_0}^{\pi_{E+I}, \gamma}(s) = \sum_{t=0}^{\infty} \gamma^t P(s_t = s | \rho_0, \pi_{E+I})$$

Note that for brevity, we write $\mathbb{E}_{s \sim d_{\rho_0}^{\pi_{E+I}, \gamma}, a \sim \pi}\left[.\right]$ as $\mathbb{E}_{\pi_{E+I}}\left[.\right]$ instead. To get rid of the dependency on samples from $\pi_{E+I}$, we use the local approximation [12, 13] shown below:

$$L_{E+I}^{\alpha}(\pi_{E+I}) = \alpha J_E(\pi_E) + \sum_{s \in \mathcal{S}} d_{\rho_0}^{\pi_E, \gamma}(s) \sum_{a \in \mathcal{A}} \pi_{E+I}(a|s) U_{\max}^{\pi_E}(s_t, a_t). \tag{17}$$

The discounted state visitation frequency of $\pi_{E+I}$ is replaced with that of $\pi_E$. This local approximation is useful because if we can find a $\pi_0$ such that $L_{E+I}^{\alpha}(\pi_0) = J_{E+I}^{\alpha}(\pi_0)$, the local approximation matches the target in the first order: $\nabla_{\pi_{E+I}} L_{E+I}^{\alpha}(\pi_{E+I})|_{\pi_{E+I}=\pi_0} = \nabla_{\pi_{E+I}} J_{E+I}^{\alpha}(\pi_{E+I})|_{\pi_{E+I}=\pi_0}$. This implies that if $L_{E+I}^{\alpha}(\pi_{E+I})$ is improved, $J_{E+I}^{\alpha}(\pi_{E+I})|_{\pi_{E+I}=\pi_0}$ will be improved as well. Schulman et al. [12] suggested that this local approximation is valid when $\mathbb{E}_{\pi_E}\left[D_{\text{KL}}(\pi_E||\pi_{E+I})\right] \leq \delta$, where $\epsilon$ is a predefined threshold. Rewriting the objective in Equation 13 using local approximation (Equation 17) leads to the desired objective:

$$J_{E+I}^{\alpha}(\pi_{E+I}) - \alpha J_E(\pi_E) \approx L_{E+I}^{\alpha}(\pi_{E+I}) - \alpha J_E(\pi_E) \tag{18}$$

$$= \sum_{s \in \mathcal{S}} d_{\rho_0}^{\pi_E, \gamma}(s) \sum_{a \in \mathcal{A}} \pi_{E+I}(a|s) U_{\max}^{\pi_E}(s_t, a_t) \tag{19}$$

$$= \sum_{s \in \mathcal{S}} d_{\rho_0}^{\pi_E, \gamma}(s) \sum_{a \in \mathcal{A}} \frac{\pi_{E+I}(a|s)}{\pi_E(a|s)} U_{\max}^{\pi_E}(s_t, a_t) \quad \text{(Importance sampling)}$$
$$\tag{20}$$

$$= \mathbb{E}_{\pi_E}\left[\frac{\pi_{E+I}(a|s)}{\pi_E(a|s)} U_{\max}^{\pi_E}(s, a)\right] \tag{21}$$

$$\text{subject to } \mathbb{E}_{s \sim \pi_E}\left[D_{\text{KL}}(\pi_E(.|s)||\pi_{E+I}(.|s))\right] \leq \delta,$$

Note that to make use of the approximation proposed in [12, 13], we make the assumption that in the beginning of the max-stage, $\pi_E = \pi_{E+I}$. Under this assumption, $\pi_E$ serves as $\pi_0$ (see above). This enables updating $\pi_{E+I}$ using the local approximation. We leave relaxing this assumption as future work.

**Min-stage objective $U_{\min}^{\pi_{E+I}}$.** We show that the objective (LHS) can be approximated by the RHS shown below

$$\max_{\pi_E} \alpha J_E(\pi_E) - J_{E+I}^{\alpha}(\pi_{E+I}) \approx \max_{\pi_E} \mathbb{E}_{\pi_{E+I}}\left[\frac{\pi_E(a|s)}{\pi_{E+I}(a|s)} U_{\min}^{\pi_{E+I}}(s, a)\right] \tag{22}$$

$$\text{subject to } \mathbb{E}_{s \sim \pi_{E+I}}\left[D_{\text{KL}}(\pi_{E+I}(.|s)||\pi_E(.|s))\right] \leq \delta.$$

The derivation for the min-stage is quite similar to that of the max-stage. Thus we only outline the key elements:

$$\alpha J_E(\pi_E) - J_{E+I}^\alpha(\pi_{E+I}) = -J_{E+I}^\alpha(\pi_{E+I}) + \alpha J_E(\pi_E) \tag{23}$$

$$= -\mathbb{E}_{s_0}\left[V_{E+I}^{\pi_{E+I},\alpha}(s_0)\right] + \alpha\mathbb{E}_{\pi_E}\left[\sum_{t=0}^{\infty}\gamma^t r_t^E\right] \tag{24}$$

$$= \mathbb{E}_{\pi_E}\left[-V_{E+I}^{\pi_{E+I},\alpha}(s_0) + \sum_{t=0}^{\infty}\gamma^t\alpha r_t^E\right] \tag{25}$$

$$= \mathbb{E}_{\pi_E}\left[\sum_{t=0}^{\infty}\gamma^t(\alpha r_t^E + \gamma V_{E+I}^{\pi_{E+I},\alpha}(s_{t+1}) - V_{E+I}^{\pi_{E+I},\alpha}(s_t))\right] \tag{26}$$

$$= \mathbb{E}_{\pi_E}\left[\sum_{t=0}^{\infty}\gamma^t\alpha r_t^E + \gamma V_{E+I}^{\pi_{E+I},\alpha}(s_{t+1}) - V_{E+I}^{\pi_{E+I},\alpha}(s_t)\right]. \tag{27}$$

Since we empirically find that $V_{E+I}^{\pi_{E+I},\alpha}$ is hard to fit under a continually changing $\alpha$, we replace $V_{E+I}^{\pi_{E+I},\alpha}$ with $V_{E+I}^{\pi_{E+I}}$ in Equation 27, and rewrite the objective as:

$$\alpha J_E(\pi_E) - J_{E+I}^\alpha(\pi_{E+I}) \approx \mathbb{E}_{\pi_E}\left[\sum_{t=0}^{\infty}\gamma^t\alpha r_t^E + \gamma V_{E+I}^{\pi_{E+I}}(s_{t+1}) - V_{E+I}^{\pi_{E+I}}(s_t)\right] \tag{28}$$

$$= \mathbb{E}_{\pi_E}\left[\sum_{t=0}^{\infty}\gamma^t U_{\min}^{\pi_{E+I}}(s_t, a_t)\right] \tag{29}$$

$$= \mathbb{E}_{\pi_E}\left[U_{\min}^{\pi_{E+I}}(s_t, a_t)\right] \qquad\qquad \text{(Rewriting by } d_{\rho_0}^{\pi_E,\gamma}) \tag{30}$$

$$\approx \mathbb{E}_{\pi_{E+I}}\left[\frac{\pi_E(a|s)}{\pi_{E+I}(a|s)}U_{\min}^{\pi_{E+I}}(s_t, a_t)\right] \qquad\qquad \text{(See Equations 16 to 21)} \tag{31}$$

$$\text{subject to } \mathbb{E}_{s\sim\pi_{E+I}}\left[\mathrm{D}_{\mathrm{KL}}(\pi_{E+I}(.|s)||\pi_E(.|s))\right] \le \delta,$$

$\alpha$ **optimization** Let $g(\alpha) := \max_{\pi_{E+I}\in\Pi}\min_{\pi_E\in\Pi} J_{E+I}^\alpha(\pi_{E+I}) - \alpha J_E(\pi_E)$. As $\pi_E$ and $\pi_{E+I}$ are not yet optimal during the training process, we solve $\min_\alpha g(\alpha)$ using stochastic gradient descent as shown below:

$$\alpha \leftarrow \alpha - \beta\nabla_\alpha g(\alpha) \tag{32}$$

$$= \alpha - \beta\nabla_\alpha(J_{E+I}^\alpha(\pi_{E+I}) - \alpha J_E(\pi_E)) \tag{33}$$

$$= \alpha - \beta(J_E(\pi_{E+I}) - J_E(\pi_E)) \tag{34}$$

$$\approx \alpha - \beta\mathbb{E}_{\pi_E}\left[\frac{\pi_{E+I}(a|s)}{\pi_E(a|s)}A_E^{\pi_E}(s,a)\right], \tag{35}$$

$$\text{subject to } \mathbb{E}_{s\sim\pi_E}\left[\mathrm{D}_{\mathrm{KL}}(\pi_E(.|s)||\pi_{E+I}(.|s))\right] \le \delta,$$

where $\beta$ is the learning rate of $\alpha$ and $A_E^{\pi_E}(s_t, a_t) := r_t^E + \gamma V_E^{\pi_E}(s_{t+1}) - V_E^{\pi_E}(s_t)$ denotes the extrinsic advantage of $\pi_E$.

## A.2 Implementation details

### A.2.1 Algorithm

**Clipped objective** We use proximal policy optimization (PPO) [10] to optimize the constrained objectives in Equation 6 and Equation 9. The policies $\pi_E$ and $\pi_{E+I}$ are obtained by solving the following optimization problems with clipped objectives:

- **Max-stage:**

$$\max_{\pi_{E+I}}\mathbb{E}_{\pi_E}\left[\min\left\{\frac{\pi_{E+I}(a|s)}{\pi_E(a|s)}U_{\max}^{\pi_E}(s,a), \mathrm{clip}(\frac{\pi_{E+I}(a|s)}{\pi_E(a|s)}, 1-\epsilon, 1+\epsilon)U_{\max}^{\pi_E}(s,a)\right\}\right] \tag{36}$$

- **Min-stage:**

$$\max_{\pi_E} \mathbb{E}_{\pi_{E+I}} \left[ \min \left\{ \frac{\pi_E(a|s)}{\pi_{E+I}(a|s)} U_{\min}^{\pi_{E+I}}(s,a), \mathrm{clip}(\frac{\pi_E(a|s)}{\pi_{E+I}(a|s)}, 1-\epsilon, 1+\epsilon) U_{\min}^{\pi_{E+I}}(s,a) \right\} \right] \quad (37)$$

where $\epsilon$ denotes the clipping threshold for PPO. We will detail the choices of $\epsilon$ in the following paragraphs.

**Rearranging the expression for GAE**    To leverage generalized advantage estimation (GAE) [19], we rearrange $U_{\max}^{\pi_E}$ and $U_{\min}^{\pi_{E+I}}$ to relate them to the advantage functions. The advantage function $A_E^{\pi_E}$ and $A_{E+I}^{\pi_{E+I}}$ are defined as:

$$A_E^{\pi_E}(s_t) = r_t^E + \gamma V_E^{\pi_E}(s_{t+1}) - V_E^{\pi_E}(s_t) \tag{38}$$
$$A_{E+I}^{\pi_{E+I}}(s_t) = r_t^E + r_t^I + \gamma V_{E+I}^{\pi_{E+I}}(s_{t+1}) - V_{E+I}^{\pi_{E+I}}(s_t). \tag{39}$$

As such, we can rewrite $U_{\max}^{\pi_E}$ and $U_{\min}^{\pi_{E+I}}$ as:

$$U_{\max}^{\pi_E}(s_t, a_t) = (1+\alpha)r_t^E + r_t^I + \gamma \alpha V_E^{\pi_E}(s_{t+1}) - \alpha V_E^{\pi_E}(s_t) \tag{40}$$
$$= r_t^E + r_t^I + \alpha A_E^{\pi_E}(s_t) \tag{41}$$
$$U_{\min}^{\pi_{E+I}}(s_t, a_t) = \alpha r_t^E + \gamma V_{E+I}^{\pi_{E+I}}(s_{t+1}) - V_{E+I}^{\pi_{E+I}}(s_t) \tag{42}$$
$$= (\alpha - 1)r_t^E - r_t^I + A_E^{\pi_{E+I}}(s_t). \tag{43}$$

**Extrinsic reward normalization**    For each parallel worker, we maintain the running average of the extrinsic rewards $\bar{r}^E$. This value is updated in the following manner at each timestep $t$:

$$\bar{r}^E \leftarrow \gamma \bar{r}^E + r_t^E.$$

The extrinsic rewards are then rescaled by the standard deviation of $\bar{r}^E$ across workers as shown below:

$$r_t^E \leftarrow r_t^E / \mathrm{Var}\left[\bar{r}^E\right].$$

**Auxiliary objectives**    The auxiliary objectives for each stage are listed below:

- **Max-stage:** We train the extrinsic policy $\pi_E$ to maximize $J_E(\pi_E)$ using PPO as shown below:

$$\max_{\pi_E} \mathbb{E}_{\pi_E^{\mathrm{old}}} \left[ \min \left\{ \frac{\pi_E(a|s)}{\pi_E^{\mathrm{old}}(a|s)} A_E^{\pi_E^{\mathrm{old}}}(s,a), \mathrm{clip}(\frac{\pi_E(a|s)}{\pi_E^{\mathrm{old}}(a|s)}, 1-\epsilon, 1+\epsilon) A_E^{\pi_E^{\mathrm{old}}}(s,a) \right\} \right], \quad (44)$$

where $\pi_E^{\mathrm{old}}$ denotes the extrinsic policy that collects trajectories at the current iteration.

- **Min-stage:** We train the mixed policy $\pi_{E+I}$ to maximize $J_{E+I}(\pi_{E+I})$ using PPO as shown below:

$$\max_{\pi_{E+I}} \mathbb{E}_{\pi_{E+I}^{\mathrm{old}}} \left[ \min \left\{ \frac{\pi_{E+I}(a|s)}{\pi_{E+I}^{\mathrm{old}}(a|s)} A_{E+I}^{\pi_{E+I}^{\mathrm{old}}}(s,a), \mathrm{clip}(\frac{\pi_{E+I}(a|s)}{\pi_{E+I}^{\mathrm{old}}(a|s)}, 1-\epsilon, 1+\epsilon) A_{E+I}^{\pi_{E+I}^{\mathrm{old}}}(s,a) \right\} \right], \quad (45)$$

where $\pi_{E+I}^{\mathrm{old}}$ denotes the mixed policy that collects trajectories at the current iteration.

**Clipping the derivative of** $\alpha$    The derivative of $\alpha$, $\delta\alpha$ (see Section 3.3), is clipped to be within $(-\epsilon_\alpha, \epsilon_\alpha)$, where $\epsilon_\alpha$ is a non-negative constant.

**Codebase**    We implemented our method and each baseline on top of the `rlpyt`[1] codebase. We thank Adam Stooke and the `rlpyt` team for their excellent work producing this codebase.

**Summary**    We outline the steps of our method in Algorithm 2.

---

[1] https://github.com/astooke/rlpyt

**Algorithm 2** Detailed Extrinsic-Intrinsic Policy Optimization (EIPO)

1: Initialize policies $\pi_{E+I}$ and $\pi_E$ and value functions $V_{E+I}^{\pi_{E+I}}$ and $V_E^{\pi_E}$
2: Set `max_stage[0]` $\leftarrow$ `False`, and $J[0] \leftarrow 0$
3: **for** $i = 1 \cdots$ **do**           $\triangleright$ $i$ denotes iteration index
4:   **if** `max_stage[i - 1]` **then**     $\triangleright$ Max-stage: rollout by $\pi_E$ and update $\pi_{E+I}$
5:    Collect trajectories $\tau_E$ using $\pi_E$
6:    Compute $U_{\max}^{\pi_E}(s_t, a_t) \, \forall (s_t, a_t) \in \tau_E$ using Eq. 40
7:    Update $\pi_{E+I}$ by Eq. 36 and $\pi_E$ by Eq. 45
8:    Update $V_E^{\pi_E}$ (see [19])
9:    $J[i] \leftarrow J_{E+I}^{\alpha}(\pi_{E+I}) - \alpha J_E(\pi_E)$
10:    `max_stage[i]` $\leftarrow J[i] - J[i-1] \leq 0$
11:   **else**           $\triangleright$ Min-stage: rollout by $\pi_{E+I}$ and update $\pi_E$
12:    Collect trajectories $\tau_{E+I}$ using $\pi_{E+I}$
13:    Compute $U_{\min}^{\pi_{E+I}}(s_t, a_t) \, \forall (s_t, a_t) \in \tau_{E+I}$ using Eq. 40
14:    Update $\pi_E$ by Eq. 36 and $\pi_{E+I}$ by Eq. 44
15:    Update $V_{E+I}^{\pi_{E+I}}$ (see [19])
16:    $J[i] \leftarrow J_{E+I}^{\alpha}(\pi_{E+I}) - \alpha J_E(\pi_E)$
17:    `max_stage[i]` $\leftarrow J[i] - J[i-1] \geq 0$
18:   **end if**
19:   **if** `max_stage[i - 1]` = `True` and `max_stage[i]` = `False` **then**
20:    Update $\alpha$ (Eq. 32)      $\triangleright$ Update when the max-stage is done
21:   **end if**
22: **end for**

## A.2.2 Models

**Network architecture** Let `Conv2D(ic, oc, k, s, p)` be a 2D convolutional neural network layer with `ic` input channels, `oc` output channels, kernel size `k`, stride size `s`, and padding `p`. Let `LSTM(n, m)` and `MLP(n, m)` be a long-short term memory layer and a multi-layer perceptron (MLP) with `n`-dimensional inputs and `m`-dimensional outputs, respectively.

For policies and value functions, the CNN backbone is implemented as two CNN layers, `Conv2D(1, 16, 8, 4, 0)` and `Conv2D(16, 32, 4, 2, 1)`, followed by an LSTM layer, `LSTM($CNN_OUTPUT_SIZE, 512)`. The policies $\pi_{E+I}$ and $\pi_{E+I}$, and the value functions $V_{E+I}^{\pi_{E+I}}$ and $V_{E+I}^{\pi_{E+I}}$ have separate MLPs that take the LSTM outputs as inputs. Each policy MLP is `MLP(512,` $|\mathcal{A}|)$, and each value function MLP is `MLP(512, 1)`.

For the prediction networks and target networks in *RND*, we use a model architecture with three CNN layers followed by three MLP layers. The CNN layers are defined as follows: `Conv2D(1, 32, 8, 4, 0)`, `Conv2D(32, 64, 4, 2, 0)`, and `Conv2D(64, 64, 3, 1, 0)`, with `LeakyReLU` activations in between each layer. The MLP layers are defined as follows: `MLP(7*7*64, 512)`, `MLP(512, 512)`, and `MLP(512, 512)`, with `ReLU` activations in between each layer.

## A.2.3 Baselines

- *Decay-RND (DY)*: We propose a variant of Ext-norm-RND where intrinsic rewards are progressively scaled down to eliminate exploration bias over time. Intrinsic rewards $r_t^I$ are scaled by $\lambda(i)$, where $i$ denotes the iteration number. The objective function turns into $J_{E+I} = \mathbb{E}_\pi \left[ \sum_{t=0}^{\infty} \gamma^t (r_t^E + \lambda(i) r_t^I) \right]$, where $\lambda(i)$ is defined as $\lambda(i) = \text{clip}(\frac{i}{I}(\lambda_{\max} - \lambda_{\min}), \lambda_{\min}, \lambda_{\max})$, where $\lambda_{\max}$ and $\lambda_{\min}$ denote the predefined maximum and minimum $\lambda(i)$, and $I$ is the iteration after which decay is fixed. In all of our experiments, we split the entire training process into 3000 iterations with equal number of frames and set $\lambda_{\max} = 1$ and $\lambda_{\min} = 0.00001$ and $I = 3000$.

- *Decoupled-RND (DC) [17]*: We adapt the method proposed in [17] to Ext-norm-RND. Two policies $\pi_{E+I}$ and $\pi_E$ are trained as follows:

$$\pi_{E+I}^* = \arg\max_{\pi_{E+I}} \mathbb{E}_{\pi_{E+I}} \left[ \sum_{t=0}^{\infty} \gamma^t (r_t^E + r_t^I) - D_{\text{KL}}(\pi_E || \pi_{E+I}) \right], \quad \pi_E^* = \arg\max_{\pi_E} \mathbb{E}_{\pi_I} \left[ \sum_{t=0}^{\infty} \gamma^t r_t^E \right].$$

Table 2: PPO Hyperparameters

| Name | Value |
|---|---|
| Num. parallel workers | 128 |
| Num. minibatches of PPO | 4 |
| Trajectory length of each worker | 128 |
| Learning rate of policy/value function | 0.0001 |
| Discount $\gamma$ | 0.99 |
| Value loss weight | 1.0 |
| Gradient norm bound | 1.0 |
| GAE $\lambda$ | 0.95 |
| Num. PPO epochs | 4 |
| Clipping ratio | 0.1 |
| Entropy loss weight | 0.001 |
| Max episode steps | 27000 |

Table 3: RND Hyperparameters

| Name | Value |
|---|---|
| Drop probability | 0.25 |
| Intrinsic reward scaling $\lambda$ | 1.0 |
| Learning rate | 0.0001 |

The exploration policy $\pi_{E+I}$ collects trajectories for training $\pi_{E+I}$ and $\pi_E$. $\pi_{E+I}$ and $\pi_E$ maximize mixed and extrinsic objectives, respectively. The $D_{KL}(\pi_E||\pi_{E+I})$ term in the objective incentivizes $\pi_I$ to perform differently from $\pi_E$. We train both $\pi_{E+I}$ and $\pi_E$ using PPO. In addition to policies, we train value functions $V_{E+I}^{\pi_{E+I}}$ and $V_E^{\pi_E}$. Both policies and value functions share the same CNN backbone.

**Hyperparameters** The hyperparameters for PPO, RND, and EIPO are listed in Table 2, Table 3, and Table 4, respectively.

## A.3 Environment details

**Pycolab**

- State space $\mathcal{S}$: $\mathbb{R}^{3\times84\times84}$, $5 \times 5$ cropped top-down view of the agent's surroundings, scaled to an $84 \times 84$ RGB image (see the code in the supplementary materials for details).

- Action space $\mathcal{A}$: $\{$UP, DOWN, LEFT, RIGHT, NO ACTION$\}$.

- Extrinsic reward function $\mathcal{R}_E$: See section 4.1.

**Atari**

- State space $\mathcal{S}$: $\mathbb{R}^{1\times84\times84}$, $84 \times 84$ gray images.

- Action space $\mathcal{A}$: Depends on the environment.

- Extrinsic reward function $\mathcal{R}_E$: Depends on the environment.

Table 4: EIPO Hyperparameters

| Name | Value |
|---|---|
| Initial $\alpha$ | 0.5 |
| Step size $\beta$ of $\alpha$ | 0.005 |
| Clipping range of $\delta\alpha$ $(-\epsilon_\alpha, \epsilon_\alpha)$ | 0.05 |

Table 5: Tuned $\lambda$ value for each environment

| | $r^E \ll \lambda r^I$ | $r^E < \lambda r^I$ | $r^E \approx \lambda r^I$ | $r^E > \lambda r^I$ | $r^E \gg \lambda r^I$ |
|---|---|---|---|---|---|
| Enduro | 38800 | 600 | 388 | 50 | 0.1 |
| Jamesbond | 2000 | 50 | 23 | 0.25 | 0.1 |
| StarGunner | 600 | 15 | 6.33 | 0.1 | 0.05 |
| TimePilot | 500 | 15 | 5 | 0.25 | 0.1 |
| YarsRevenge | 3000 | 50 | 30 | 5 | 0.1 |
| Venture | 500 | 50 | 5 | 0.5 | 0.05 |

## A.4 Evaluation details

### A.4.1 Probability of improvement

We validate whether EIPO prevents the possible performance degradation introduced by intrinsic rewards, and consistently either improves or matches the performance of PPO in 61 Atari games. As our goal is to investigate if an algorithm generally performs better than PPO instead of the performance gain, we evaluate each algorithm using the "probability of improvement" metric suggested in [18]. We ran at least 5 random seeds for each method in each environment, collecting the median extrinsic returns within the last 100 episodes and calculating the probability of improvements $P(X \geq \text{PPO})$ [2] with 95%-confidence interval against PPO for each algorithm $X$. The confidence interval is estimated using the bootstrapping method. The probability of improvement is defined as:

$$P(X \geq Y) = \frac{1}{N^2} \sum_{i=1}^N \sum_{j=1}^N S(x_i, y_j), \quad S(x_i, y_j) = \begin{cases} 1, x_i \geq y_j \\ 0, x_i < y_j, \end{cases}$$

where $x_i$ and $y_j$ denote the samples of median of extrinsic return trials of algorithms $X$ and $Y$, respectively.

We also define strict probability of improvement to measure how an algorithm dominate others:

$$P(X > Y) = \frac{1}{N^2} \sum_{i=1}^N \sum_{j=1}^N S(x_i, y_j), \quad S(x_i, y_j) = \begin{cases} 1, x_i > y_j \\ \frac{1}{2}, x_i = y_j \\ 0, x_i < y_j, \end{cases}$$

### A.4.2 Normalized score

In addition, we report the PPO-normalized score [18] to validate whether EIPO preserves the performance gain granted by RND when applicable. Let $p_X$ be the distribution of median extrinsic returns over the last 100 episodes of training for an algorithm $X$. Defining $p_{\text{PPO}}$ as the distribution of mean extrinsic returns in the last 100 episodes of training for PPO, and $p_{\text{rand}}$ as the average extrinsic return of a random policy, then the PPO-normalized score of algorithm $X$ is defined as: $\frac{p_X - p_{\text{rand}}}{p_{\text{PPO}} - p_{\text{rand}}}$.

### A.4.3 $\lambda$ tuning

Table 5 lists the $\lambda$ values used in Section 4.5.

## A.5 RND-dominating games

Table A.5 shows that the mean and median PPO-normalized score of each method with 95%-confidence interval in the set of games where RND performs better than PPO.

The set of games where RND performs better than PPO are listed below:

- AirRaid

- Alien

---

[2]Note that Agarwal et al. [18] define probability of improvements as $P(X > Y)$ while we adapt it to $P(X \geq Y)$ as we measure the likelihood an algorithm $X$ can match or exceed an algorithm $Y$.

Table 6: EIPO exhibits higher performance gains than RND in the games where RND is better than PPO. Despite being slightly below RND in terms of median score, EIPO attains the highest median among baselines other than RND.

| Algorithm | PPO-normalized score | | | |
|---|---|---|---|---|
| | Mean (CI) | | Median (CI) | |
| RND | 384.57 | (85.57, 756.69) | **1.22** | (1.17, 1.26) |
| Ext-norm RND | 427.08 | (86.53, 851.52) | 1.05 | (1.02, 1.14) |
| Decay-RND | 383.83 | (84.19, 753.17) | 1.04 | (1.01, 1.11) |
| Decoupled-RND | 1.54 | (1.09, 2.12) | 1.00 | (0.96, 1.06) |
| EPIO-RND | **435.56** | (109.45, 874.88) | 1.13 | (1.06, 1.23) |

- Assault
- Asteroids
- BankHeist
- Berzerk
- Bowling
- Boxing
- Breakout
- Carnival
- Centipede
- ChopperCommand
- DemonAttack
- DoubleDunk
- FishingDerby
- Frostbite
- Gopher
- Hero
- Kangaroo
- KungFuMaster
- MontezumaRevenge
- MsPacman
- Phoenix
- Pooyan
- Riverraid
- RoadRunner
- SpaceInvaders
- Tutankham
- UpNDown
- Venture

## A.6 Scores for each Atari game

The mean scores for each method on all Atari games are presented in Table 7.

## A.7 Complete learning curves

We present the learning curves of each method in Figure 6, and the evolution of $\alpha$ in EIPO in Figure 7 on all Atari games.

|  | PPO | RND | Ext-norm RND | Decay-RND | Decouple-RND | Ours |
|---|---|---|---|---|---|---|
| Adventure | **0.0** | 0.0 | 0.0 | 0.0 | 0.0 | 0.0 |
| AirRaid | 34693.2 | 42219.9 | 36462.4 | 36444.7 | 30356.4 | **50418.2** |
| Alien | 1891.0 | 2434.9 | 2152.1 | 2148.3 | 2386.9 | **2536.7** |
| Amidar | **1053.4** | 1037.0 | 736.4 | 909.5 | 987.1 | 901.3 |
| Assault | 8131.9 | 10592.2 | **10985.1** | 9504.3 | 8404.5 | 10771.1 |
| Asterix | 14313.0 | 14112.9 | 16872.5 | **20078.0** | 11292.2 | 12471.8 |
| Asteroids | 1360.9 | 1431.1 | **1433.8** | 1385.0 | 1426.7 | 1389.4 |
| BankHeist | 1336.3 | 1345.1 | 1339.0 | **1346.0** | 1334.8 | 1333.2 |
| BattleZone | 83826.0 | 47128.0 | 72117.0 | 61939.0 | 59461.7 | **87478.0** |
| BeamRider | 7278.7 | 7085.1 | 7460.0 | 7802.5 | 7215.4 | **7854.6** |
| Berzerk | 1113.8 | **1478.5** | 1459.0 | 1455.9 | 1196.4 | 1426.6 |
| Bowling | 17.4 | 14.6 | 26.0 | 32.6 | 19.0 | **52.3** |
| Boxing | 79.5 | 79.9 | **79.9** | 60.3 | 1.9 | 79.5 |
| Breakout | 565.7 | **658.6** | 570.6 | 545.7 | 479.3 | 529.5 |
| Carnival | 5019.3 | 5052.9 | 4513.4 | 4790.8 | 4964.7 | **5534.3** |
| Centipede | 5938.2 | 6444.4 | 6832.3 | **6860.0** | 6675.3 | 6460.8 |
| ChopperCommand | 8225.1 | **9465.9** | 8629.8 | 8559.0 | 6649.7 | 8008.4 |
| CrazyClimber | **151202.6** | 147676.5 | 135970.3 | 140333.9 | 138956.7 | 137036.7 |
| DemonAttack | 5678.8 | 7070.2 | 9039.0 | 6707.0 | 8990.1 | **9984.4** |
| DoubleDunk | -1.3 | **18.0** | -1.1 | -1.0 | -1.0 | -1.9 |
| ElevatorAction | 45703.7 | 9777.6 | 12121.4 | 19250.5 | 42557.3 | **48303.7** |
| Enduro | 1024.7 | 797.5 | 815.0 | **1095.9** | 677.7 | 1092.6 |
| FishingDerby | 35.3 | **47.8** | 28.9 | 36.3 | 36.7 | 37.5 |
| Freeway | 31.1 | 25.8 | 33.4 | **33.4** | 33.1 | 33.3 |
| Frostbite | 1011.3 | 3445.3 | 1731.4 | 3368.2 | 2115.2 | **5289.6** |
| Gopher | 5544.2 | **13035.8** | 2859.6 | 11034.9 | 9964.6 | 4928.8 |
| Gravitar | 1682.2 | 1089.8 | 1874.1 | 1437.0 | 1253.4 | **1921.1** |
| Hero | 29883.7 | **36850.3** | 26781.2 | 29842.4 | 33889.1 | 36101.3 |
| IceHockey | 6.0 | 4.4 | 8.7 | 6.9 | 9.9 | **10.4** |
| Jamesbond | 13415.9 | 3971.6 | 13474.4 | 12322.4 | 14995.6 | **15352.0** |
| JourneyEscape | -429.7 | -1035.0 | -663.7 | -413.2 | -327.8 | **-309.3** |
| Kaboom | **1883.5** | 1592.5 | 1866.6 | 1860.8 | 1830.7 | 1852.3 |
| Kangaroo | 6092.4 | 8058.9 | 8293.4 | 9361.9 | **12043.3** | 10150.8 |
| Krull | 9874.1 | 8199.4 | 9921.4 | 9832.0 | 9551.3 | **10006.2** |
| KungFuMaster | 47266.5 | **66954.2** | 48944.5 | 47403.2 | 45666.8 | 48329.4 |
| MontezumaRevenge | 0.2 | 2280.0 | **2500.0** | 2217.0 | 0.0 | 2485.0 |
| MsPacman | 4996.9 | **5326.6** | 5289.7 | 4792.5 | 4325.0 | 4767.4 |
| NameThisGame | 11127.7 | 10596.1 | 10300.7 | 11831.5 | **11918.0** | 11294.9 |
| Phoenix | 8265.0 | 10537.9 | 10922.9 | 11494.5 | **17960.8** | 16344.1 |
| Pitfall | **0.0** | -2.7 | -6.1 | -0.6 | -1.5 | -0.3 |
| Pong | 20.9 | 20.9 | **20.9** | 20.9 | 20.9 | 20.9 |
| Pooyan | 5773.4 | **7535.8** | 5508.7 | 5430.9 | 4834.7 | 5924.6 |
| PrivateEye | 97.5 | 86.0 | **114.9** | 98.8 | 99.7 | 99.5 |
| Qbert | **23863.8** | 16530.9 | 22387.8 | 22443.3 | 22289.5 | 22750.7 |
| Riverraid | 10231.3 | 11073.6 | 11700.4 | 13365.7 | 13285.1 | **14978.4** |
| RoadRunner | 45922.6 | 46518.4 | **58777.7** | 44684.2 | 42694.3 | 58708.8 |
| Robotank | 37.4 | 24.9 | 38.5 | 40.1 | 40.7 | **40.9** |
| Seaquest | 1453.9 | 1128.6 | **1986.0** | 1426.6 | 1821.5 | 1838.3 |
| Skiing | -12243.3 | -14780.8 | -11594.8 | -11093.5 | **-8986.6** | -9238.4 |
| Solaris | 2357.7 | 2006.5 | 2120.9 | 2251.7 | **2751.0** | 2572.0 |
| SpaceInvaders | 1621.0 | **1871.4** | 1495.3 | 1692.0 | 1375.7 | 1637.6 |
| StarGunner | 21036.0 | 16394.9 | 16884.7 | 32325.8 | 42299.5 | **50798.5** |
| Tennis | -0.1 | -4.7 | **4.6** | -0.1 | -8.2 | -0.1 |
| TimePilot | 19544.5 | 9180.5 | **21409.4** | 20034.2 | 19223.8 | 21039.8 |
| Tutankham | 199.9 | **235.3** | 230.6 | 214.0 | 216.1 | 231.8 |
| UpNDown | 276884.8 | **317426.2** | 310520.6 | 266774.5 | 290323.4 | 294218.8 |
| Venture | 102.1 | 1149.7 | 1348.6 | **1451.8** | 1438.8 | 1146.3 |
| VideoPinball | 360562.5 | 327741.8 | 350534.3 | **406508.8** | 389578.5 | 392005.7 |
| WizardOfWor | 11912.8 | 9580.3 | 11845.2 | 11751.7 | 10732.7 | **12512.8** |
| YarsRevenge | 92555.9 | 73411.4 | 85851.9 | 77850.0 | 124983.6 | **149710.8** |
| Zaxxon | 14418.2 | 11801.9 | 11779.6 | 15085.5 | **16813.3** | 12713.3 |

Table 7: The mean scores of each method in 61 Atari games.

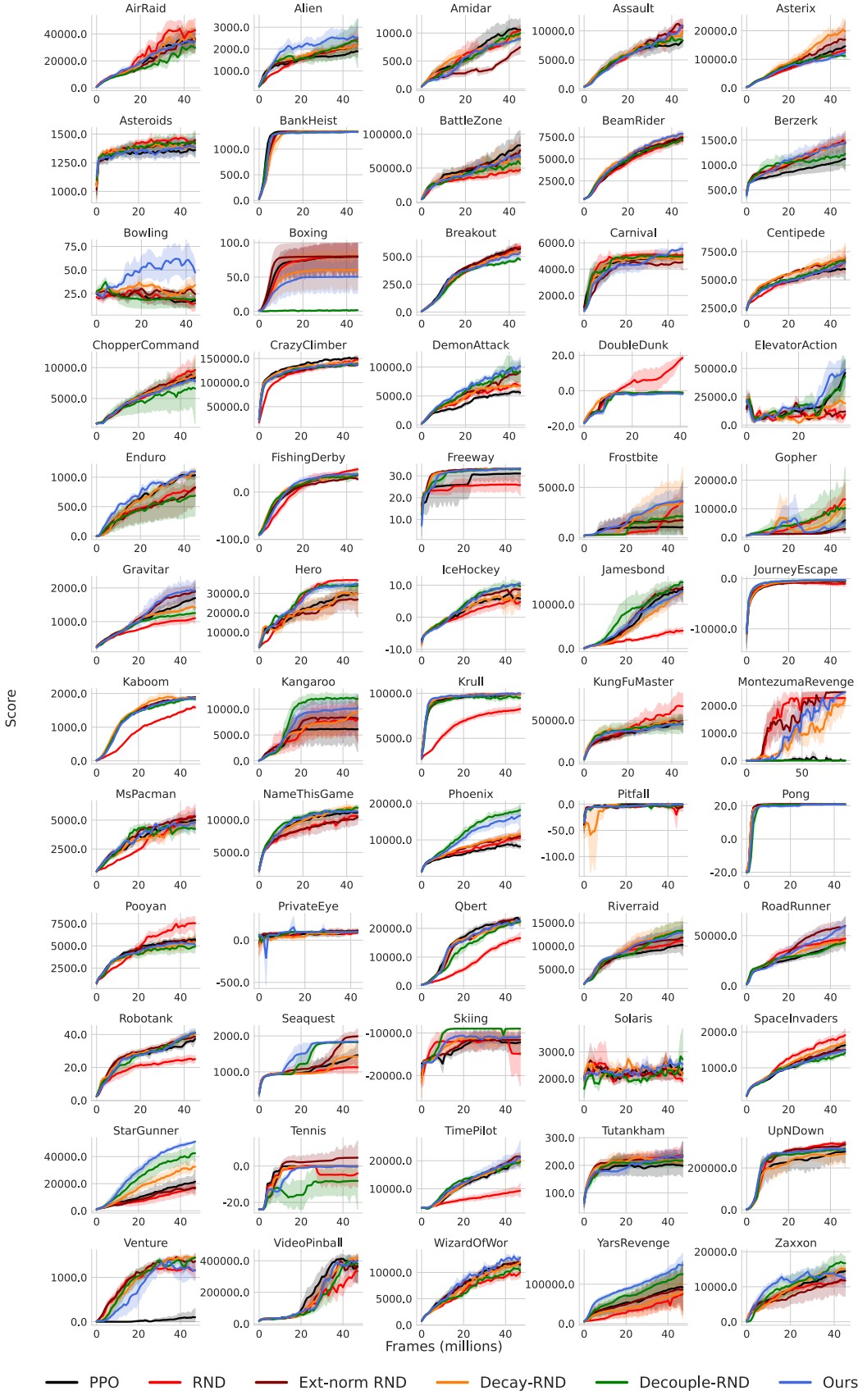

Figure 6: Game score for each baseline on 60 Atari games. Each curve represents the average score across at least 5 random seeds. In all games, we either match or outperform PPO. In a large majority of games, we either match or outperform RND. In a handful of games, our method does significantly better than both PPO and RND (*Star Gunner*, *Bowling*, *Yars Revenge*, *Phoenix*, *Seaquest*).

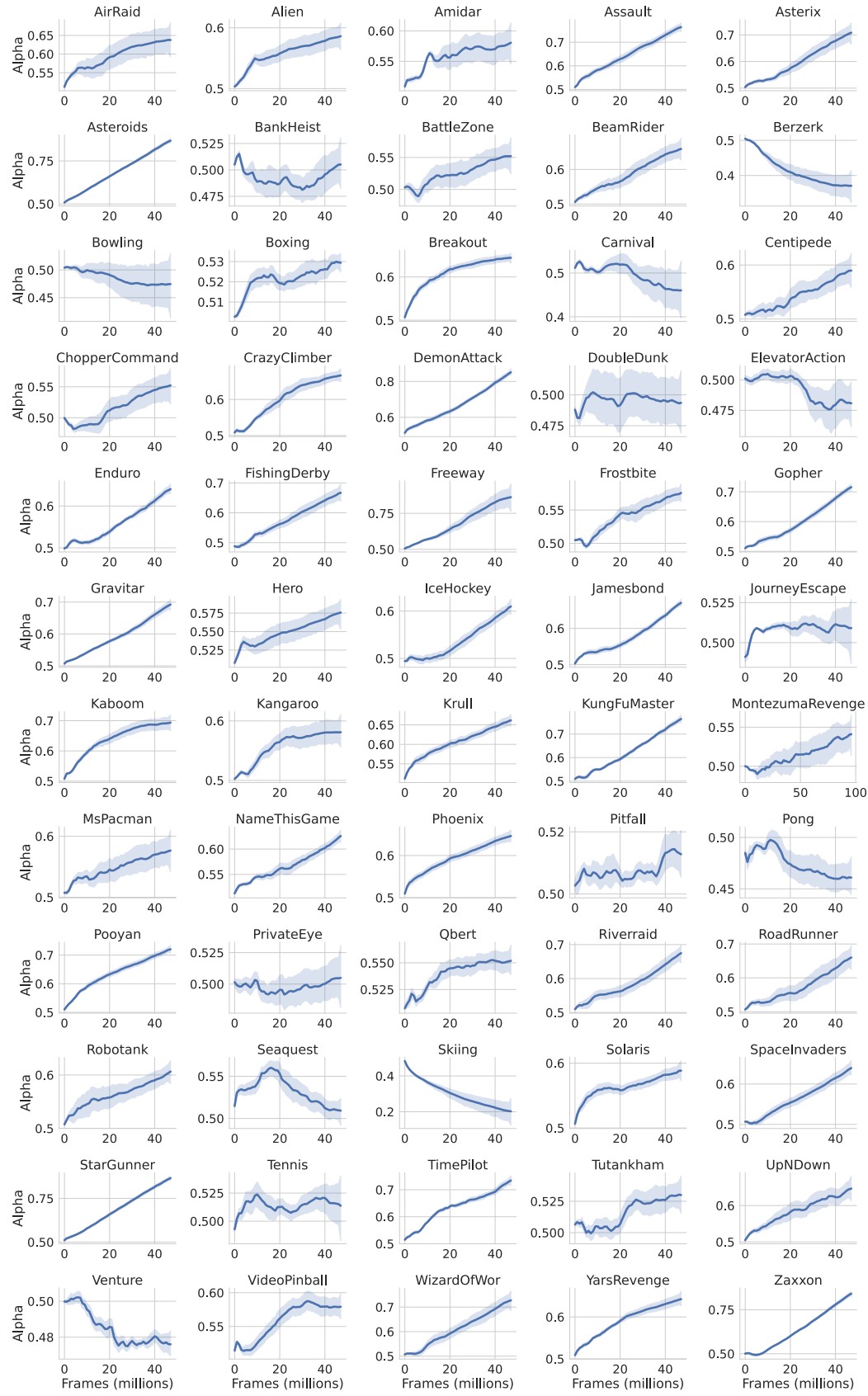

Figure 7: The evolution of $\alpha$ in EIPO on all 61 Atari environments. The variance in $\alpha$ trajectories across environments supports the hypothesis that decaying the intrinsic reward is difficult to hand-tune, and may not always be the best strategy.

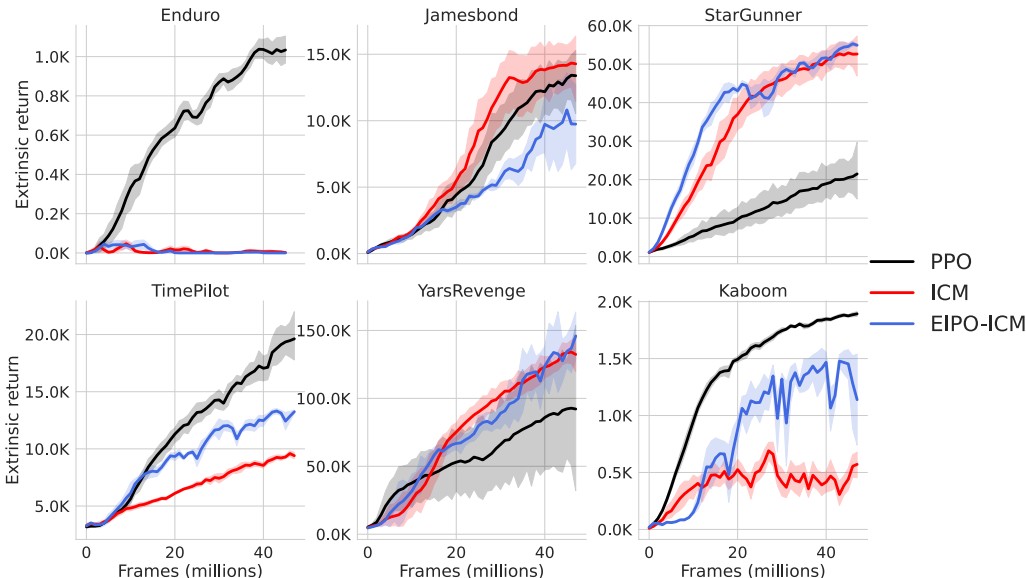

Figure 8: EIPO-ICM successfully matches ICM when it outperforms PPO, and closes the gap with PPO when ICM underperforms. In *Kaboom*, the screen flashes a rapid sequence of bright colors when the agent dies, causing ICM to generate high intrinsic reward at these states. Even in such games where the intrinsic and extrinsic reward signals are misaligned, our method is able to close the performance gap. In extreme cases where the intrinsic and extrinsic rewards are steeply misaligned (*Enduro*), our methods inability to completely turn off the effects of intrinsic rewards results in subpar performance. On the same environment however, we see that RND does perform well (Fig. 6). This supports our view that extending our method to optimize between different intrinsic reward signals as well as intrinsic and extrinsic rewards could be an interesting direction for future work.

## A.8 ICM

In addition to RND, we test our method on ICM [6] - another popular bonus-based exploration method. The learning curves on 6 Atari environments can be seen in Fig. 8.

## A.9 Related Work

Our work is related to the paradigm of reward design. Meriçli et al. [20] uses genetic programming to optimize the reward function for robot soccer. Sorg et al. [21] learns a reward function for planning via gradient ascent on the expected return of a tree search planning algorithm (e.g., Monte Carlo Tree Search). Guo et al. [22] extends [21] using deep learning, improving tree search performance in Atari games. The work [23] learns a reward function to improve the performance of model-free RL algorithms by performing policy gradient updates on the reward function. Zheng et al. [24] takes a meta-learning approach to learn a reward function that improves an RL algorithm's sample efficiency in unseen environments. Hu et al. [25] learns a weighting function that scales the given shaping rewards [26] at each state and action. These lines of work are complimentary to EIPO, which is agnostic to the choice of intrinsic reward and could be used in tandem with a learned reward function.