# OpenReview forum: "Redeeming intrinsic rewards via constrained optimization"
_NeurIPS.cc/2022/Conference — NeurIPS 2022 Accept_

### Official Review · Reviewer_JLyn · 2022-07-08

**Rating:** 8
**Confidence:** 3
**Soundness:** 4 excellent
**Presentation:** 4 excellent
**Contribution:** 3 good

**Summary:**

Difficult exploration tasks which rely on sparse rewards can be particularly challenging for traditional RL methods. Injecting intrinsic reward functions can increase reward density, but also introduces bias that can hinder performance in more tractable exploration tasks. Regulating this bias often reduces to a balancing act between intrinsic vs extrinsic reward influence using manual hyperparameter tuning. This paper proposes an objective function that explicitly minimizes the difference between the learned, combined intrinsic + extrinsic policy and an optimal policy under only the extrinsic reward. The objective function is reformulated using lagrangian duality and the policy learning is constrained using TRPO to provide tractability. The method is then evaluated on 61 Atari exploration tasks that vary in difficulty. Results show that the proposed method can match state of the art performance on difficult tasks while still achieving PPO-level performance on simpler tasks, all without manual tuning of the reward balance.

**Questions:**

I have no major questions or suggestions, other than the minor suggestions provided above.

**Limitations:**

I would have liked to see more discussion around the limitations of this approach, or possible applications where this may not be appropriate.

**Strengths And Weaknesses:**

## Originality

The core concept is simple but elegant.  Formulation of the minimization constraint between mixed- and extrinsic-trained policies, reformulation using langrange duals, and the use of a TRPO-like objective to improve trainability were all creative. Overall, I like that the idea is approachable but very effective.

## Quality

The overall quality of the paper is excellent, with a few minor caveats:

- I thought the abstract was a little weak. It's more pronounced because the rest of the paper is very solid, so the abstract feels thin by comparison. A few suggestions:
	- Some may not be familiar with "intrinsic rewards," so maybe highlight the link with reward sparsity.
	- Maybe include a few more details about your approach beyond just "constrained policy optimization"
- Very minor, but each word in your title should be capitalized except for "via".

On the positive:

- As mentioned above, the core concept is solid and well-presented.
- Detailing of the methodology was excellent, from preliminaries through implementation details.
- The number of experiments and the detail of the analysis were great.

## Clarity

The paper was clearly written and approachable. The description of the problem and the formulation of the solution follow a coherent narrative and are straightforward to understand.

## Significance

I don't feel like anything in this work is particularly ground breaking, but rather an elegant solution to a specific problem. That said, I do appreciate the breadth of evaluation and it will be interesting to see this approach applied to more real-world scenarios.

---

> ### Author Response · Authors · 2022-08-02
> **response**
>
> We’re happy to hear that you enjoyed the breadth of the empirical results, and found the core concept and methodology compelling.
>
> > I thought the abstract was a little weak. Some may not be familiar with "intrinsic rewards," so maybe highlight the link with reward sparsity. Maybe include a few more details about your approach beyond just "constrained policy optimization". Very minor, but each word in your title should be capitalized except for "via".
> >
>
> **Answer:** Thanks for pointing it out. We appreciate your feedback concerning the abstract. We have updated the abstract with an explanation of intrinsic rewards and reward sparsity, and more details on our approach. We have capitalized each word in the title as well. We are happy to modify the abstract further based on the reviewer’s feedback.
>
> > I don't feel like anything in this work is particularly ground breaking, but rather an elegant solution to a specific problem. That said, I do appreciate the breadth of evaluation and it will be interesting to see this approach applied to more real-world scenarios.
> >
>
> **Answer:**
>
> We are glad the reviewer finds that our method provides an elegant solution to a problem. While we agree that we have demonstrated results in the specific context of balancing intrinsic rewards against extrinsic rewards, we would like to position our contribution in context of its broader significance:
>
> Exploration-exploitation is among the core challenges in RL. Prior to our work, when measured across Atari games, the standard exploration strategy for “on-policy” policy gradient algorithms was to randomly sample actions from a gaussian or boltzmann policy $\pi(a|s)$. While intrinsic reward formulations like RND and ICM improved performance on a few sparse reward games, when measured on average across all ATARI games, they were not found to be better than naive random sampling strategies (i.e., they improved performance on some tasks and led to worse performance on others). Due to inconsistent performance across tasks and the need for extensive tuning of intrinsic against extrinsic reward, methods like RND/ICM have not found widespread use by RL practitioners. To the best of our knowledge, EIPO provides the first demonstration of a more sophisticated exploration bonus (i.e, RND) outperforming a naive exploration strategy without the need for manual tuning. As such, we believe it has the potential of widespread adoption as a core-component of on-policy policy gradient algorithms such as PPO. At the same time, we agree with the reviewer that it would be interesting to see the performance of our method on real-world tasks.
>
> Another way to look at our work is: A major problem in RL is reward design. Practitioners often spend a substantial amount of time tuning the relative weighting of different reward terms. Intrinsic rewards can be seen as a “special case” of reward shaping where EIPO has been demonstrated to be successful. Our results suggest that EIPO may provide an elegant solution to the more general problem of reward shaping — something we are very excited to explore in the future.
>
> > I would have liked to see more discussion around the limitations of this approach, or possible applications where this may not be appropriate.
> >
>
> **Answer:** Thanks for asking about the limitations. In our view these are:
>
> - EIPO is dependent on a good choice of an intrinsic reward function (such as RND). We show that our method performs well using two state-of-the-art intrinsic reward functions: RND and ICM (see Appendix A.8). However neither of these intrinsic reward functions have a theoretical justification, nor are they guaranteed to fully explore the state space. In such cases, EIPO wouldn’t be able to overcome the inherent limitation of the intrinsic reward function itself. Coming up with “good” intrinsic reward functions remains an open challenge. Furthermore, EIPO makes the assumption that the policy $\pi_{E+I}$ is close to $\pi_{E}$ in the optimization, which we were able to successfully leverage with state-of-the-art intrinsic rewards. Although unlikely, it’s possible that with some intrinsic reward, the assumptions made by EIPO are violated.
> - While we have presented strong empirical results, we are still working on a theoretical analysis of convergence of our algorithm. We believe such an analysis will further boost confidence that our method can scale to more complex real-world tasks.
>
> We have included a note of the limitations of EIPO in the revised version of our paper.
>
> In summary, based on empirical evidence so far, we believe EIPO along with RND, can be used instead of PPO with naive exploration strategies such as $\epsilon$-greedy, wherever practitioners use PPO today.

---

### Official Review · Reviewer_sfNt · 2022-07-10

**Rating:** 7
**Confidence:** 5
**Soundness:** 4 excellent
**Presentation:** 4 excellent
**Contribution:** 4 excellent

**Summary:**

The authors pose the problem of exploration using intrinsic motivation as one of constrained optimization.
Naturally, the goal of the agent maximizing the intrinsic+extrinsic reward should be to perform AT LEAST as good as an agent simply maximizing the intrinsic reward.

As such, they present a constraint where the E+I agent needs to perform at least as good as the E agent when measured only on E (extrinsic) rewards.

Empirical evidence shows this method indeed works.

**Questions:**

My main question is around section 3.3 (and 3.2)

Why collect data using pi_{E+I} when optimizing pi_{E}? Why not optimize pi_{E} when using data collected by pi_{E}?
Why not optimize both agents in parallel at each time? Is there a restriction against optimizing pi_{E} and pi_{E+I} in both max and min stages, just making sure to use the appropriate importance sampling ratios?

**Limitations:**

-

**Strengths And Weaknesses:**

Strengths:

- Motivation is clear. The approach is solid.
- Empirical results look very good.
- Paper is overall well written.

Weaknesses:

- Wouldn't expect strong theoretical justification, but overall this is an optimization objective with 3 parts that can be presented as a 2-3 time-scale stochastic optimization scheme and convergence can be analyzed. Convergence analysis might shed some light on section 3.3 which seems a bit hard to comprehend.

---

> ### Author Response · Authors · 2022-08-02
> **response**
>
> We’re pleased to find that the reviewer found the empirical results to be very good, and that our motivation and approach are clear and well presented. We clarify the reviewer’s concerns below.
>
> -----
> > Wouldn't expect strong theoretical justification, but overall this is an optimization objective with 3 parts that can be presented as a 2-3 time-scale stochastic optimization scheme and convergence can be analyzed. Convergence analysis might shed some light on section 3.3 which seems a bit hard to comprehend.
> >
>
> **Answer:**
>
> Thanks for raising this point. The convergence analysis requires two parts:
>
> **Part 1: Does the stationary point solution of the optimization problem in Eq. 5 remove the bias of intrinsic rewards from the mixed policy $\pi_{E+I}$? In other words, do $\pi_{E+I}$ and $\pi_{E}$ achieve the same expected return when we optimize $\pi_{E+I}$ using EIPO?**
>
> Our optimization objective is expressed as follows:
> $$
> \begin{align}
> &\min_{\alpha}\max_{\pi_{E+I}} \min_{\pi_E}  {L}(\pi_{E+I}, \pi_E, \alpha) \\\\
> {L}(\pi_{E+I}, \pi_E, \alpha)  &= \mathbb{E}_{\pi_\{E+I\}}\Big [\sum^\infty_\{t=0\} \gamma^t ((1+\alpha)R_E(s_t,a_t) + R_I(s_t, a_t))\Big] - \alpha \mathbb{E}_\{\pi_E\} \Big[ \sum^\infty_\{t=0\} \gamma^t  R_E(s_t,a_t) \Big]
> \end{align}
> $$
>
> One of the necessary conditions of the stationary solution to the above optimization problem is $\partial \mathcal{L} / \partial \alpha = 0$. Satisfying this condition implies the following identity:
>
> $$
> \begin{align}
> 0 &=\dfrac{\partial L}{\partial\alpha} =  \mathbb{E}_{\pi_\{E+I\}}\Big[\sum^\infty_\{t=0\} \gamma^t R_E(s_t,a_t) \Big] - \mathbb{E}_\{\pi_E\}\Big[\sum^\infty_\{t=0\} \gamma^t R_E(s_t,a_t) \Big] \\\\
> &\implies \mathbb{E}_\{\pi_\{E+I\}\}\Big[\sum^\infty_\{t=0\} \gamma^t R_E(s_t,a_t) \Big] = \mathbb{E}_\{\pi_E\}\Big[\sum^\infty_\{t=0\} \gamma^t R_E(s_t,a_t) \Big]
> \end{align}
> $$
>
> As the expected extrinsic returns of $\pi_{E+I}$ and $\pi_{E}$ are equal at some stationary point $\alpha^*$, it means that at the optimal point, the mixed policy $\pi_{E+I}$ is not biased by intrinsic rewards.
>
> **Part 2: Can our algorithm reach the stationary solution of Eq. 5?**
>
> As non-convexity of Eq. 5 complicates the convergence analysis, we are still working on the analysis of our alternating algorithm for solving the min-max problem, and we will update our analysis in the next version of the manuscript.
>
> We hope that our analysis takes a first step in clarifying the reviewer’s question, and we are happy to address any follow-up questions.
>
>
> -----
> > Why not optimize both agents in parallel at each time?
> >
>
> **Answer:** We indeed train both policies in parallel. However, we only use one policy to collect data and we update both policies with the same data for sample efficiency. The policy for collecting data is switched when the improvement in the optimization objective of the current data-collecting policy plateaus. For a more detailed description, pleas see Line 5 and Line 10 in Algorithm 1, as well as Section 3.3.
>
>
> -----
> > Why collect data using pi_{E+I} when optimizing pi_{E}? Why not optimize pi_{E} when using data collected by pi_{E}?
> >
>
> **Answer:** Thanks for your question. Lets consider the min-stage $\min_{\pi_E} J^\alpha_\{E+I\}(\pi_\{E+I\}) - \alpha J_E(\pi_E)$. As pointed out by the reviewer, it is definitely possible to train $\pi_{E}$ using trajectories $\tau_{E
> }$ collected from $\pi_{E}$. However, for data efficiency, if we want to simultaneously train $\pi_{E+I}$ with $\tau_{E}$, one would require either off-policy importance weight corrections on the trajectories [1] or the state density function of policies [2]. It is well known that the trajectory correction terms increase in variance with longer horizon leading to training instabilities. On the other hand, state density correction requires estimating state densities, which is known to be difficult in practice with high-dimensional state observations such as images.
>
> We can overcome these problems by making an approximations and the results described in Equations 6-8. The consequence is that we can train both $\pi_{E+I}$ and $\pi_{E}$ using $\tau_{E+I}$ using the correction ratio $\dfrac{\pi_{E}(a|s)}{\pi_{E+I}(a|s)}$ (irrespective of the problem horizon) which is easy to compute.  This is the reason why we train both $\pi_{E}$ and $\pi_{E+I}$ using $\tau_{E}$ in the min-stage. Similarly, we train $\pi_{E}$ and $\pi_{E+I}$ using $\tau_{E+I}$ in the max stage.
>
> [1] Notes on Importance Sampling and Policy Gradient ([https://nanjiang.cs.illinois.edu/files/cs598/note6.pdf](https://nanjiang.cs.illinois.edu/files/cs598/note6.pdf))
>
> [2] Liu, Yao, et al. "Off-policy policy gradient with state distribution correction."  **(2019).

---

> > ### Comment · Reviewer_sfNt · 2022-08-08
> > **Thanks**
> >
> > Thanks for your response.
> >
> > I must have missed those explanations in the paper. Indeed it sufficiently covers my concerns.
> > Overall I liked this work, very intuitive and simple scheme.

---

### Official Review · Reviewer_bbkw · 2022-07-11

**Rating:** 7
**Confidence:** 3
**Soundness:** 3 good
**Presentation:** 3 good
**Contribution:** 3 good

**Summary:**

This paper focuses on the exploration-exploitation trade-off in reinforcement learning. In particular, the authors aim to devise a method that automatically adjusts the importance of the intrinsic reward component (exploitation) and the extrinsic reward component (exploration). To this end, they pose an (extrinsic optimality) constrained policy optimization. Then, they write the Lagrangian dual problem and solve it iteratively. For implementation purposes, they leverage tools from TRPO/PPO literature. The proposed EIPO method is compatible with any intrinsic reward method; for experiments, they use RND. Finally, they conducted an extensive empirical investigation of their proposed method on all 61 Atari games.

**Questions:**

Question:

How does EIPO relate with (or compare against) the line of works in [Zheng et al. 2018; Hu et al. 2020] that also consider automatic learning/adjustment of intrinsic reward component?

[Zheng et al. 2018] Zheng et al. On learning intrinsic rewards for policy gradient 376 methods. 2018.

[Hu et al. 2020] Hu et al. Learning to utilize shaping rewards: A new approach of reward shaping. 2020.

**Limitations:**

Their proposed method does not seem to cause any direct potential negative societal impacts.

**Strengths And Weaknesses:**

Strengths:

The proposed method is interesting and novel. The existing intrinsic reward-based exploration methods (e.g., RND) need to heavily tune the parameter $\lambda$ in $r_E + \lambda r_I$ for each environment to perform better. EIPO alleviates this manual tuning effort and automatically adjusts the importance scale $\alpha$ between the intrinsic and extrinsic rewards via optimization. Further, EIPO is compatible with any intrinsic reward method. In this work, the authors do not directly learn or optimize the intrinsic reward $r_I$.

The paper is overall well written; the motivation and the story are well conveyed. It was easy to follow.

On the empirical side, the authors have done a large-scale study on all 61 Atari games and reported the results using rigorous metrics as prescribed in [18]. In particular, the empirical results validate the following: (i) the probability of improvement EIPO over PPO is higher than the baselines (variants of RND), (ii) EIPO performs better than all the baselines in the majority of games (as well as in terms of strict probability of improvement), (iii) EIPO performs comparable/better than heavily $\lambda$-tuned RND (for each environment). These empirical findings make the paper strong.

****

Weakness:

The paper is missing a detailed discussion of related work.

---

> ### Author Response · Authors · 2022-08-02
> **response**
>
> We are glad that the reviewer found our paper to be interesting, novel, well-written and with strong empirical results.
>
> > How does EIPO relate with (or compare against) the line of works in [Zheng et al. 2018; Hu et al. 2020] that also consider automatic learning/adjustment of intrinsic reward component?
> [Zheng et al. 2018] Zheng et al. On learning intrinsic rewards for policy gradient methods. 2018.
> [Hu et al. 2020] Hu et al. Learning to utilize shaping rewards: A new approach of reward shaping. 2020.
> >
>
> **Answer:**
>
> Thanks for pointing to these references. The primary focus of Zheng et al. 2018 and Hu et al. 2020 is on learning an intrinsic reward function. EIPO is agnostic to the choice of intrinsic reward. It can be used with the intrinsic rewards proposed by Zheng et al. 2018 and Hu et al. 2020., or any other intrinsic reward formulation such as ICM or RND that we used in the paper. We select RND because it achieves state-of-the-art performance in hard exploration Atari games. Furthermore, RND is easier to implement than Zheng et al. 2018 and Hu et al. 2020, and has an open source implementation.
>
> We have included a discussion of these works in our related work section in Appendix A.9 (we will fit this section into the main paper during the next revision).
>
> Please let us know if we can answer any other questions or provide further clarifications.

---

> > ### Comment · Reviewer_bbkw · 2022-08-07
> > **Thanks for the response**
> >
> > I thank the authors for their response and for clarifying my question.

---

### Meta-Review · Area_Chair_ZQUU · 2022-08-22

**Recommendation:** Accept
**Confidence:** Certain

**Metareview:**

Balancing between extrinsic rewards and intrinsic rewards is an important challenge for exploration in RL. This paper proposes a simple yet effective way to automatically adjust the balance between them. The large-scale empirical result across 61 Atari games shows a strong improvement over the baseline approaches. All of the reviewers agreed that the proposed method is novel, and the empirical results are convincing. The reviewers had no major concern about the paper. Thus, I recommend accepting this paper.

**Award:**

No

---

### Decision · Program_Chairs · 2022-09-14

Accept